# An Instrumental Value for Data Production and its Application to Data Pricing

**Rui Ai** [1]   **Boxiang Lyu** [2]   **Zhaoran Wang** [3]   **Zhuoran Yang** [4]   **Haifeng Xu** [5]

## Abstract

We develop a framework for capturing the *instrumental value* of data production processes, which accounts for two key factors: (a) the *context* of the agent's decision-making; (b) how much data or information the buyer already possesses. We "micro-found" our data valuation function by establishing its connection to classic notions of signals and information design in economics. When instantiated in Bayesian linear regression, our value naturally corresponds to information gain. Applying our proposed data value in Bayesian linear regression for monopoly pricing, we show that if the seller can fully customize data production, she can extract the first-best revenue (i.e., full surplus) from *any* population of buyers, i.e., achieving *first-degree price discrimination*. If data can only be constructed from an existing data pool, this limits the seller's ability to customize, and achieving first-best revenue becomes generally impossible. However, we design a mechanism that achieves seller revenue at most $\log(\kappa)$ less than the first-best, where $\kappa$ is the condition number associated with the data matrix. As a corollary, the seller extracts the first-best revenue in the *multi-armed bandits* special case.

## 1. Introduction

Trading data forms an ever-growing segment in today's digital economy. Advances in machine learning motivate companies to apply data-driven approaches to solve a growing variety of problems, ranging from personalized market- ing (Goldsmith & Freiden, 2004) to data-hungry applications such as natural language processing (Kang et al., 2020) and self-driving (Ni et al., 2020). These applications highlight the demand for high-quality labeled data, a demand that a growing number of data sellers wish to meet. Start-up companies such as Scale AI, Parallel Domain, Dataloop, and V7 compete with established companies, such as Acxiom, Oracle, and Nielsen, in this growing arena, underlying the importance of formal analyses of buying and selling data.

Like physical goods and systems, data has both instrumental value and intrinsic value at the same time. For example, the New York Times (NYT) as a news source has its own intrinsic value, containing major up-to-date news. However, serving as a Corpus for a particular machine learning task like retrieval augmented generation (RAG) (Lewis et al., 2020), its value towards improving the performance of the task is instrumental. The fundamental difference between instrumental vs. intrinsic value is widely studied in philosophy. Citing the widely accepted definitions from Beardsley (1965): *X has instrumental value* means *X is conducive to something that has intrinsic value*. In the above example of NYT data, its instrumental value to a RAG task depends on the extra values it adds and, crucially, depends on what data the buyer already has. For instance, when the RAG system already has the data from other substitutable news sources (e.g., the Washington Post, Wall Street Journal, etc.), the NYT data's instrumental value will become smaller, despite holding the same intrinsic value. This is also the reason that most economic activities, including the sale of data, are based on items' instrumental values, as illustrated by Foster (1981): "The instrumental efficiency of the economic process is the criterion of judgment in terms of which, and only in terms of which, we may resolve economic problems". To our knowledge, there is no notion of such instrumental data value in current literature; this is what we embark on studying in this work.

Data sold by commercial companies falls into two situations, which we dub *perfect customization* and *limited customization*. Perfect customization can offer data labels for any feature directions specified by the buyer, whereas limited customization can only curate data based on some pre-collected dataset, hence has limited flexibility. For example, a company that wishes to develop a model for image classification can either **[perfect customization]** submit its

---

[1]Institute for Data, Systems, and Society, MIT, Cambridge, MA, USA [2]Booth School of Business, University of Chicago, Chicago, IL, USA [3]Department of Industrial Engineering and Management Sciences, Northwestern University, Evanston, IL, USA [4]Department of Statistics and Data Science, Yale University, New Haven, CT, USA [5]Department of Computer Science, University of Chicago, Chicago, IL, USA. Correspondence to: Rui Ai <ruiai@mit.edu>, Haifeng Xu <haifengxu@uchicago.edu>.

*Proceedings of the 42nd International Conference on Machine Learning*, Vancouver, Canada, PMLR 267, 2025. Copyright 2025 by the author(s).

most desirable set of unlabelled images and buy labels for these images from the data seller, or **[limited customization]** purchase a labeled dataset of useful, though possibly not the most desirable images.

### 1.1. Our Contributions

In the paper, we first characterize valid valuations in Sections 2 and 3 and then present a concrete application, say, mechanism design for data pricing, in Section 4. We sketch our contributions as follows.

**A new notion for data's instrumental value.** We first introduce a general framework to quantify the instrumental value of data which closely relates to a generalized notion of Bregman Divergence, inspired by Frankel & Kamenica (2019). Our notion rests on a deep duality relation between the measure of parameter uncertainty and the measure of data value. We show that, when instantiated in Bayesian linear regression, our value corresponds precisely to the natural notion of information gain of the new data, relative to existing data. This special form of data instrumental value offers a useful basis for practitioners since the Bayesian linear model is a widely studied foundational model, and has been proven to be a good approximation when we do not have specific knowledge about the underlying model structure. Quoting Besbes & Zeevi (2015), *linearity offers the greatest robustness under misspecification.* Following this spirit, we use this particular valuation form to demonstrate our insights. Notably, our instrumental data value crucially differs from Data Shapley (Ghorbani & Zou, 2019) and its variants. Data Shapley captures a data point's *expected* contribution to an ML task, whereas our data value captures a dataset's *marginal* contribution beyond existing data, hence more naturally captures the dataset's economic value. Another strong advantage is that our new data valuation can be computed with only one model retraining, hence much more practical to compute in reality (see more detailed discussions in Section 2.3).

**The power of perfect data customization.** With the introduced measure of data instrumental value, we next turn to its application to selling data. When the seller can customize data production for buyers, we show that there exists a pricing mechanism that achieves first-best revenue; that is, it extracts full welfare, hence leaves no surplus for buyers. This result highlights a fundamental difference between selling data and selling physical goods. Specifically, when selling goods, the power of customization is limited, at most through randomized allocation, leading to buyers' utility linearity (in allocation probabilities). However, for selling data, there is significantly more power of customization by producing various (high-dimensional and non-linear) derivatives of data. Such rich "data allocation" space is the fundamental reason for the first-best revenue extraction. It

also hints at the potential concerns of highly-screwed (towards seller) surplus distribution on data markets and calls for future government regulation.

**Approximate optimality for selling data under limited customization.** When the seller has limited customization ability and can only curate data derivatives from a pre-collected dataset, we exhibit a novel data-selling mechanism based on the SVD decomposition of the data matrix and show that it almost extracts full welfare, up to a small constant which relates to the condition number of the data matrix of the to-be-sold dataset. Therefore, under this mechanism, buyers can only enjoy at most a constant amount of surplus.

### 1.2. Related Works

Our work is closely related to the literature on *valuing data and information* (Ghorbani & Zou, 2019; Jia et al., 2019; Frankel & Kamenica, 2019; Schoch et al., 2022), and *selling data and information* (Admati & Pfleiderer, 1988; Babaioff et al., 2012; Xiang & Sarvary, 2013; Bergemann & Bonatti, 2015; Hörner & Skrzypacz, 2016; Kastl et al., 2018; Chen et al., 2019; Segura-Rodriguez, 2021; Liu et al., 2021a; Chen et al., 2022). Due to the space limit, we discuss these and more related works thoroughly in Appendix A.1.

## 2. An Instrumental Value of Data Production and its Micro-foundation

In this section, we introduce a new way to quantify the instrumental value of a *Data Production Process* (DPP), which captures a DPP's expected value for a certain task. Its formulation is fundamentally different from the widely studied *Data Shapley* (Roth, 1988; Ghorbani & Zou, 2019; Jia et al., 2019), which quantifies the value of a realized dataset, hence can usually be evaluated only by accessing the realized data. In contrast, our developed value for DPPs can be evaluated with just the knowledge about how the data are produced, but *without* the need to truly access the realized data. Additionally, the developed valuation function is *relative* to prior information as well as the user's context, hence it is possible in our definition that high-quality data may have low value to certain user if the user already has sufficient prior knowledge or if the user cares about a context that is not reflected in the data.

Our new data valuation function is rooted in a few classic subjects, including Bayesian regression, decision-making and the economic value of information, which we now elaborate on.

### 2.1. The Context: Bayesian Regression and Data Production

Our valuation function for DPP is based on the context of a fundamental *statistical* learning problem — i.e., estimating parameters of a generic regression problem $y = f_\beta(x) + \epsilon$, where $f_\beta(\cdot)$ is a function parameterized by $\beta$. Here, $x \in \mathbb{R}^d$ is the feature vector and $\epsilon$ is some zero-mean random noise, capturing the measuring error. Let tuple $(x, y)$ denote a generic data point whereas $\{(x_i, y_i)\}_{i=1}^n$ denote a dataset with $n$ points. A core challenge in studying the value of a data production procedure[1] is that the value needs to be well-defined not only for the process of simply producing data records $\{(x_i, y_i)\}_{i=1}^n$, but also for any *post-processing* of the data. For instance, instead of directly giving away all data records $\{(x_i, y_i)\}_{i=1}^n$, the seller could also just produce a single averaged feature $\bar{x} = \frac{1}{n}\sum_{i=1}^n x_i$ and its corresponding label $\bar{y} = \frac{1}{n}\sum_{i=1}^n y_i$. More general statistics could include weighted data combinations, some moments of the data, or even beyond, which all carry information about the underlying parameter $\beta$. This motivates us to consider the following general notion of a *data production process*.

**Definition 2.1** (Data Production Process (DPP)). A data production process (DPP) is described by a data generating distribution $g_\beta(D)$, which produces realized data $D$ with probability $g_\beta(D)$ under parameter $\beta$. When $\beta$ is clear from context, we simply use $g$ to denote this DPP.

*Example* 1 (Examples of DPPs). Consider a linear model $y = \langle x, \beta \rangle + \epsilon$ with uninformative Gaussian prior $\beta \sim \mathcal{N}(0, I_2)$. A learner would like to purchase data to predict $\langle x, \beta \rangle$ for $x = (1, 1)$. A simple example of DPPs — described by two feature directions $x_1 = (1, 3), x_2 = (3, 1)$ — simply produces response variables via model $y = \langle x, \beta \rangle + \epsilon$ along feature direction $x_1, x_2$: for example, $y_1 = -1$ and $y_2 = 1$. In this case, $x_1, x_2$ specified a DPP whereas the realized $y_1, y_2$ are the data $D$ with data generating distribution $g_\beta(\{y_1, y_2\}) = \mathcal{E}(y_1 - \langle x_1, \beta \rangle)\mathcal{E}(y_2 - \langle x_2, \beta \rangle)$, where $\mathcal{E}(\cdot)$ is the noise distribution.

Besides directly producing data $y_1, y_2$ for directions $x_1, x_2$, one could also produce the data $\bar{y}$ for feature direction $\bar{x} = \frac{x_1 + x_2}{2}$ using DPP, which illustrates the rich space of DPPs. Perhaps more interestingly, we could also design a DPP that produces data $y^\perp = y_1 - y_2$ for direction $x^\perp = x_1 - x_2$. This DPP is of no interest to the learner above since $x^\perp$ is orthogonal to the learner's interested direction $x$. Nevertheless, the realized $y^\perp$ does carry information about $\beta$ and hence may be of interest to other users with a different context $x'$. Such careful curation of DPPs is a

salient feature of data sale that is intrinsically different from classic pricing problems of physical goods (Myerson, 1981). As we show later, it gives data sellers the power to tailor their data production to be useful only for one particular buyer but of little use to others, hence increasing sellers' power of *price discrimination*.

In principle, a DPP $g$ can be an arbitrary data production process and does not even need to be related to the underlying regression model $y = f_\beta(x) + \epsilon$ so long as it carries information about $\beta$ (e.g., a fully informative DPP could even directly reveal $\beta$). However, throughout this paper and similar to Example 1, we will mostly think of $g$ as being curated from certain data records $\{(x_i, y_i)\}_{i=1}^n$, for example, either as the full or partial list of $\{(x_i, y_i)\}_{i=1}^n$ or as some statistics (e.g., mean, variance, moments, etc.) computed using these data records. This more realistically captures how data is generated and stored in most ML problems, and also is more intuitive to think about. In these situations, the randomness of the data production process $g$ inherits from the model's noise $\epsilon$ and possibly extra randomness the data owner may add (see Example 4 for more details). Our valuation will be able to capture the value of all these variants. Throughout the paper, we assume the data-generating distribution $g_\beta(D)$ is publicly known (though $\beta$ is unknown); the DPPs in Example 1 are all described by varied feature directions or their functions.

As in standard Bayesian regression, we assume the data buyer possesses a prior distribution $q$ over the parameter $\beta$ and aims to further improve his prior $q$ by purchasing additional data. In reality, $q$ can be a strong prior estimated from much data that the buyer already has, or can be a weak prior as an uninformative distribution. In either situation, additional data can help to refine the prior $q$, forming the posterior $p$. Hence, a data buyer can use the realized data $D$ produced by DPP $g$ to update his belief about $\beta$ as follows[2]

$$\text{Distribution of data: } \mathbb{P}(D) = \int_\beta q(\beta) \cdot g_\beta(D)d\beta = g \circ q \tag{1}$$

Posterior updates from realized data $D$:

$$p(\beta \,|\, q, D) \propto q(\beta) \cdot g_\beta(D) \text{ for each } \beta. \tag{2}$$

For instance, how the data producing distribution $g_\beta(D)$ and posterior update $p(\beta \,|\, q, D)$ lead to variance reduction in Example 1 can be calculated as follows.

*Example* 2 (Example 1 Continued). If we produce data $\bar{y}$, the variance of the to-be-predicted variable $\langle x, \beta \rangle$ reduces from 2 to $\frac{2}{17}$, same as fully revealing original data records $\{(x_1, y_1), (x_2, y_2)\}$. However, if we only reveal $(x^\perp, y^\perp)$, the variance remains 2. Notably, these improvements can

---

[1]Data production has the same meaning as data generation or curation. We choose to use the term "production" mainly to emphasize the *active* generation of data, especially for the economic purpose of data sales.

[2]This is also known as the convolution of the parameter distribution $q(\beta)$ and model $g_\beta$.

all be calculated without knowing the realized data. This will be useful for selling DPPs since it means the value of a DPP can be quantified without seeing the realized data. This resolves the concern that data will lose its value after being seen, say the seller can reveal $\bar{x}$ or $x^{\perp}$ for advertisement.

For brevity, we refer to the posterior as $p$ when $q, D$ are clear from the context. We sometimes consider the situation with null data production, i.e., $g = \emptyset$, where no Bayes update happens and $p(\cdot \,|\, q, \emptyset) = q$.

Finally, for our theory, we need a natural generalization of Bregman divergence to *concave functionals* over (continuous) distributions in order to accommodate the continuous parameter space for $\beta$ (Cilingir et al., 2020).

**Definition 2.2** (Concave Functionals and Generalized Bregman Divergence (cf. Definition B.1)). We say *functional* $F(\cdot)$ is concave if for any $\lambda \in [0, 1]$, we have

$$F(\lambda p + (1 - \lambda)q) \geq \lambda F(p) + (1 - \lambda)F(q).$$

Moreover, we say $\nabla F(\cdot)$ is its (functional) superdifferential if

$$F(q) + \langle \nabla F(q), p - q \rangle \geq F(p),$$

for any probability distributions $p$ and $q$. The generalized Bregman divergence induced by functional $F$ is defined as

$$D_F(p, q) = F(q) - F(p) + \langle \nabla F(q), p - q \rangle.$$

## 2.2. The Value for a Data Production Process and its Micro-Foundation

Throughout, we assume the Bayesian regression model $y = f_\beta(x) + \epsilon$ and any designed DPP $g$ are publicly known. Our goal is to develop and characterize a class of valuation function that captures the instrumental value of $g$ to a particular task.

Our proposed valuation is rooted in a fundamental microeconomic problem of Bayesian decision-making (BDM). Recall that a standard BDM problem is described by a utility function $u : \mathcal{A} \times \mathcal{Y} \to \mathbb{R}$ where $u(a, y)$ is the utility under *action* $a \in \mathcal{A}$ and a random *state of the world* $y \in \mathcal{Y}$ that captures uncertainty in the decision making. As an example, in a simple production decision problem, $u(a, y)$ could be $ay - a^2$ where $a$ is the number of products to be produced, $y$ is the price of the products (often a random variable that needs to be predicted), and $a^2$ captures the production costs for producing $a$ amount. Our following definition of a contextual variant of the above problem simply says that the random variable $y$ can be context-dependent.

**Definition 2.3** (Contextual Bayesian Decision Making (CBDM)). A CBDM problem is a tuple $\langle u, y[x] = f_\beta(x) + \epsilon \rangle$ where the utility function $u : \mathcal{A} \times \mathcal{Y} \to \mathbb{R}$ represents a standard BDM problem and $y[x]$ follows a distribution specified by *context* $x$ under response model $y = f_\beta(x) + \epsilon$.

Note that CBDM simply enriches standard BDM by allowing the decision to depend on some particular context $x$, which then allows the decision maker to have a more fine-grained distribution estimation $y[x]$ of the random state $y$. For instance, still in the above production decision problem, the context $x$ could be the demographic feature of a certain population, and $y[x]$ is the to-be-predicted price targeting this particular population. Similarly, we may predict $\mathbb{E}[y[x]] = f_\beta(x)$ for some other tasks in an average manner.

Knowing the context $x$, the data buyer can improve his decision-making by purchasing data produced by DPP $g$ to refine his estimation of $\beta$ (hence refine the prediction of $y$) from his prior belief $q(\beta)$ to posterior $p(\beta|D, q)$. In a fully Bayesian world, such data should always increase the decision maker's utility, and this utility increase in expectation is a natural candidate for the value of the DPP $g$ for the given CBDM problem. This motivates us to study the following valuation function of data.

**Definition 2.4** (Valuation Functions of a DPP). Suppose the regression model $y = f_\beta(x) + \epsilon$ and data production process $g_\beta(D)$ are public. Then a valuation function for realized data $D$ has format $\texttt{val}(D; q, x)$ that depends on the buyer's prior belief $q$ of $\beta$ and the decision context $x$.

Consequently, $\texttt{V}(g; q, x) = \mathbb{E}_{D \sim g \circ q} \texttt{val}(D; q, x)$ is called the *instrumental valuation function* for DPP $g$. Moreover, we call $\texttt{cost}(q, x) = \texttt{val}(\vee; q, x)$ the *cost of uncertainty* where $g = \vee$ denotes the fully informative DPP that directly produces data $D = \beta$ (hence before seeing the data, the belief about the to-be-seen $D$'s distribution is the prior $q = g \circ q$).

A few remarks are worthwhile to mention about Definition 2.4. For simplicity, we wish to minimize the valuation functions $\texttt{val}$'s dependence on various quantities. However, its dependence on $q, x$ is essential. First, $\texttt{val}$'s dependence on prior $q$ is natural because what the buyer originally knows affects the value of newly acquired data. More valuable data are those that significantly shift the buyer's prior, whereas the data that does not change it much is less valuable. Second, the dependence on the decision context $x$ is also natural since if the data isn't informative to the particular decision context $x$, it won't have much value for the buyer regardless of how informative it may be to estimate the directions of $\beta$ that is not useful for improving prediction of $f_\beta(x)$.[3] Finally, the $\texttt{cost}$ function is simply the value of the most informative DPP, which helps the buyer to pin down parameter $\beta$ precisely so that the only uncertainty in his decision-making is the inevitable noise $\epsilon$ from nature.

Now that we propose a format of the valuation function

---

[3]For instance, if $f_\beta(x) = \langle x, \beta \rangle$, then accurate estimation of $\beta$ alone any subspace orthogonal to $x$ will not be useful for the buyer's decision-making with context $x$.

$V(g; q, x)$. However, obviously, not all such functions should be considered "valid" valuation functions. For instance, negative valued ones or those which decrease as $g$ become more informative are clearly not reasonable candidates for data valuation. Hence, the true scientific question is what kind of functions $V(g; q, x)$ could be considered as "valid" data valuation functions. To answer this question, we resort to Bayesian decision-making in order to "microfound" valid $V(g; q, x)$s.

**Definition 2.5.** [**Valid** Value Functions and Cost of Uncertainty] A valuation function $V(g; q, x) = \mathbb{E}_{D \sim g \circ q} \text{val}(D; q, x)$ is *valid* if and only if its corresponding $\text{val}(D; q, x)$ function can be expressed as follows for some contextual Bayesian decision-making problem $\langle u, y[x] = f_\beta(x) + \epsilon \rangle$:

$$\text{val}(D; q, x) = \mathbb{E}_{\beta \sim p(\cdot|q, D) \to y \sim f_\beta(x) + \epsilon} [u(a^*(p); y) - u(a^*(q); y)],$$

(3)

where $\beta \sim p(\cdot|q, D) \to y \sim f_\beta(x) + \epsilon$ denote a process of producing $y$ by first drawing $\beta \sim p$ and then drawing $y \sim f_\beta(x) + \epsilon$; $a^*(q)$ is the optimal action $a$ when the decision maker's distribution belief about parameter $\beta$ is $q$, or formally *optimal action under parameter distribution q:* $a^*(q) = \arg\max_{a \in \mathcal{A}} \mathbb{E}_{\beta \sim q \to y \sim f_\beta(x) + \epsilon} u(a, y)$.

Analogously, we can define a coupled valid *cost of uncertainty* function, and we postpone it to Appendix B.2 for interested readers.

In other words, a valuation function $V(g; q, x)$ is *valid* if there exists a CBDM problem $\langle u, y[x] = f_\beta(x) + \epsilon \rangle$ that "microfounds" it in the sense that its corresponding $\text{val}(D; q, x)$ can be written as the utility difference between the more informed (by extra data $D$) optimal action $a^*(p)$ and original uninformed action $a^*(q)$, in expectation over the randomness of the state $y = f_\beta(x) + \epsilon$ where parameter $\beta$ is generated by posterior $p$. In a fully Bayesian world, this is precisely how much the realized data $D$ helps to increase the decision maker's expected utility.

Definition 2.5 captures valid data valuation functions from a micro-economic perspective. However, these definitions are not very helpful for us to verify whether any given $V, \text{val}, \text{cost}$ functions are valid since it may be generally difficult to uncover the underlying CBDM problem $\langle u, y[x] = f_\beta(x) + \epsilon \rangle$. Next, we offer some results that offer alternative yet *equivalent* characterizations of these functions as well as their connections. These characterizations are properties of these functions themselves, hence are much easier to verify. We leave analogous proposition for $\text{cost}(\cdot, \cdot)$ to Proposition B.4 in Appendix B.2.

**Proposition 2.6** (Characterization of Valid Valuation Functions (see Proposition B.3 for details))**.** *A valuation function* $\text{val}$ *is valid if and only if it satisfies the following properties simultaneously: (1) **No value for null data**, (2) **Positivity***

*and (3) **Invariance to data acquisition orders**.*[4]

Our first main result establishes a connection between $\text{val}$ (hence also $V$) and $\text{cost}$.

**Theorem 2.7.** *A valid data valuation* $\text{val}(D; q, x)$ *and cost of uncertainty* $\text{cost}(q, x)$ *are coupled if and only if* $\text{val}$ *is a generalized Bregman divergence (Definition 2.2) of* $\text{cost}$ *in the following sense*

$$\text{val}(D; q, x) = \text{cost}(q, x) - \text{cost}(p, x) + \langle \nabla \text{cost}(q, x), p - q \rangle,$$

*where* $p(\cdot|q, D)$ *defined in Equation (2) is the posterior of* $\beta$ *and* $\nabla \text{cost}(\cdot)$ *any superdifferential of* $\text{cost}$ *defined in Definition 2.2.*

The theorem highlights the fundamentality of *Bregman divergence*, a commonly used concept in statistical learning and online learning (Banerjee et al., 2005; Gutmann & Hirayama, 2011; Raskutti & Mukherjee, 2015), in the valuation of data and cost of uncertainty. That is, any valid candidate of data valuation functions must correspond to the Bregman divergence of some concave function that captures the cost of uncertainty. Notably, since machine learning training typically involves minimizing some divergence, our results to some extent justify the choice of the loss function. As we will see below, the selection of entropy reduction is not only due to its ease of optimization but also because it corresponds to a certain preference.

### 2.3. Comparisons to Data Shapley

Data Shapley (Ghorbani & Zou, 2019; Wang et al., 2024) is another common valuation method, yet often overstates valuation. Originating from cooperative game theory, Data Shapley allocates value to each datum in an existing dataset. It computes an expectation over exponentially many (hence intractable generally) possible scenarios as follows: $\phi_i \propto \sum_{S \subset D - \{i\}} \frac{\text{val}(S \cup \{i\}; q, x) - \text{val}(S; q, x)}{\binom{|D| - 1}{|S|}}$. Our previous result characterizes what valuation function $\text{val}$ is "valid". Theorem 2.7 proves that any natural $\text{val}$ is the Bregman divergence of some concave function, which also matches most choices in practice. More importantly, our valuation only relies on $\text{val}(D; q, x)$ and $\text{val}(D \cup \{i\}; q, x)$ (since it is the marginal value) hence is much easier to compute in practice since it only requires to evaluate $\text{val}(D \cup \{i\}; q, x)$ by retraining the model one more time with the additional data $i$. Moreover, our valuation method can be applied to any transformations of the dataset as well (so long as the DPP allows us to update parameters), so it could be applicable more broadly than Data Shapley.

We now illustrate our motivation for considering marginal contribution, while not averaging over all possible data coali-

---

[4]It means the total expected value of data is invariant to the order of data acquisition.

tions. Suppose a company hopes to buy a new confidential dataset to fine-tune its large language model; it only needs to consider the status before and after fine-tuning. However, Data Shapley gives some weight to the value of a virtual model trained by part of the original data and the new dataset. Considering many of these hypothetical situations will usually overestimate the instrumental value of the new dataset and twist the corresponding data pricing. This inconsistency arises from ignoring the timeline when each dataset is collected. The following example illustrates why the Shapley Value usually overestimates. In fact, in its extreme, suppose a seller resells a repetitive data/dataset $i$ which is already in the buyer's training data pool, this same dataset will nevertheless have a non-zero Shapley value, whereas it will have $0$ instrumental value in our setting as it's useless for updating the prior.

*Example* 3 (The comparison between $\mathtt{V}(\cdot; \cdot, \cdot)$ and Data Shapley). Assuming $\beta$ has a prior $\mathcal{N}(0, 1)$ and data follows $\mathcal{N}(\beta, 1)$, we hope to formulate the entropy reduction associated with the second datum which is $\frac{1}{2} \log(\frac{3}{2})$. Here, the first datum reduces the posterior variance from $1$ to $\frac{1}{2}$ and the second one reduces it from $\frac{1}{2}$ to $\frac{1}{3}$. By choosing $\mathtt{val}$ as negative entropy, $\mathtt{V} = \frac{1}{2} \log(\frac{3}{2})$ but $\phi_2 = \frac{\log 3}{4}$, showing Data Shapley overestimates valuation in this scenario.

Here, we use entropy reduction associated with KL divergence as a valid $\mathtt{val}(\cdot; \cdot, \cdot)$. KL divergence is commonly applied in training machine learning models, like VAE (Kingma, 2013) and GAN (Goodfellow et al., 2014). Nonetheless, our instrumental value is also legitimate for some other widely-used loss functions like $L_2$ loss.

Data Shapley corresponds to a uniform distribution (Ghorbani & Zou, 2019), and some paper tries other distributions like Beta distributions (Kwon & Zou, 2021) to capture various facets of data valuation. Our instrumental value can be regarded as a kind of limitation of Shapley value that we use a Dirac delta distribution, merely focusing on $S = D - \{i\}$. It precisely describes the realistic gains from new data with computational tractability and has potential applications in the data market, like valuing data used for fine-tuning. We visualize the advantages of instrumental value with some numerical experiments in Appendix B.3.

## 3. The Canonical Data Valuation Function for Linear Regression & Entropy

Section 2 introduced a general characterization for the value of data and its associated cost of uncertainty. In this section, we instantiate this framework in a fundamental Bayesian regression problem (i.e., linear regression) with perhaps the most widely used cost of uncertainty function (i.e., entropy), and develop *closed-form* characterization for the DPP valuation function $\mathtt{V}(\cdot; \cdot, \cdot)$ in this case. This focus on a linear regression problem is due to multiple reasons. First, the linear model is arguably the most fundamental model and has been proven to be very useful under uncertainties and misspecification (Besbes & Zeevi, 2015). Second, multiple recent works have shown that data's valuations are often "transferable" in the sense that when the same sets of data are evaluated under different learning models (e.g., linear regression or ML methods), their relative value order often does not change much though the absolute value may change (Schoch et al., 2022; Jia et al., 2021). Finally, in domains such as causal inference and clinical healthcare, linear models serve as effective approximations of real-world dynamics and provide valuable insights for future methodological developments. Given these, the valuation under the linear regression model serves as a useful basis for any future analysis of data valuations, hence, we term them the *canonical valuation function*.

Let us start by recalling the classic *Bayesian linear regression* problem, which assumes that label $y$ is produced by the following linear model $y = \langle x, \beta \rangle + \epsilon$, where $\epsilon \sim \mathcal{N}(0, \sigma(x)^2)$ is a zero-mean Gaussian noise.

The Bayesian linear regression framework shares a similar sentiment with Thompson sampling (Agrawal & Goyal, 2013). It not only holds high value in machine learning theory but also finds extensive practical applications in real life, like drug monitoring (Ammad-Ud-Din et al., 2017; van den Elsen et al., 2019).

Notably, here we allow the variance $\sigma(x)$ to also depend on the context $x$. We also assume $\beta$ follows a Gaussian prior $\mathcal{N}(\mu_q, \Sigma_q)$. One of the most widely adopted cost of uncertainty functions is the Shannon entropy (Shannon, 1948), or differential entropy for continuous distributions. We conveniently refer to both as "entropy" and defer its standard definition to Appendix C. It is easy to verify that entropy is a valid cost of uncertainty over $\langle x, \beta \rangle$ and measures the confidence in the estimation. One natural question, hence, is what is the associated data valuation function coupled with entropy? Our following result provides a closed-form characterization.

With the structure of Bayesian linear regression, we may write out exactly $\mathtt{V}(\cdot; \cdot, \cdot)$ in the Bayesian linear regression setting in the following theorem.

**Theorem 3.1.** *[The Canonical Valuation of DPPs] In classic Bayesian linear regression, the valuation function for any data production process $g$ coupled with Entropy is the following function:*

$$\mathtt{V}(g; q, x) = \mathop{\mathbb{E}}_{D \sim g \circ q} \left[ \frac{1}{2} \log(x^T \Sigma_q x) - \frac{1}{2} \log \left( x^T \Sigma_{p(\cdot | D, q)} x \right) \right],$$
(4)

*where $g \circ q$ is the data distribution as in Equation (1), $p$ is the posterior distribution over $\beta \in \mathbb{R}^d$ as in Equation (2),*

*and $\Sigma_p = \mathbb{E}_{\beta \sim p}(\beta - \mu_p)(\beta - \mu_p)^T$ is its covariance matrix.*

Theorem 3.1 offers a closed-form expression (coinciding with information gain) for the valuation of any data production process $g$, associated with entropy in linear regression — due to their fundamentality, we call the $\mathtt{V}(g; q, x)$ function in Equation (4) the *canonical valuation function* for DPPs.

So far, we have been working with the most general data production process $g$. Next, we're going to instantiate our characterization in Theorem 3.1 with a concrete, perhaps the most straightforward, data production process — that is, a collection of *data records* $\{(x_i, y_i)\}_{i=1}^n$ produced by the model $y = \langle x, \beta \rangle + \epsilon$. In this case, the data production procedure is completely determined by the data matrix $X \in \mathbb{R}^{n \times d}$ with its $i$-th row as $x_i^T$, hence denoted as $g^X$. The realized data is the realized response variable values $Y = (y_1, \ldots, y_n)^T$. This concrete DPP allows us to derive a closed-form valuation function for these data records.

**Corollary 3.2.** *[The Canonical Valuation of Producing Data Records] For any data records $\{(x_i, y_i)\}_{i=1}^n$ generated by the linear regression model $y = \langle x, \beta \rangle + \epsilon$, let $X \in \mathbb{R}^{n \times d}$ denote the design matrix. The canonical valuation of this DPP procedure, denoted as $g^X$, is characterized in closed form as follows: $\mathtt{V}(g^X; q, x) = \frac{1}{2} \log(x^T \Sigma_q x) - \frac{1}{2} \log(x^T (\Sigma_q^{-1} + X^T \Sigma^{-1} X)^{-1} x)$, where $\Sigma = diag\{\sigma(x_1)^2, \ldots, \sigma(x_n)^2\}$.*

## 4. Optimal Pricing of Data Production under Canonical Valuations

In this section, we use the developed closed-form characterization for canonical valuations of DPPs in Section 3 to study a natural optimal pricing problem for selling DPPs. While the choice of valuation functions generally may be domain-dependent, the goal of this section is to showcase a natural economic application of the newly developed valuation functions of DPPs. We study how to sell data production processes (DPPs) to a machine learner (henceforth, the *buyer*).

We consider the following standard mechanism design framework, albeit being instantiated in our DPP pricing setup with fundamentally different utility functions and allocation rules. Specifically, the buyer would like to estimate $\mathbb{E}[y[x]] = \langle x, \beta \rangle$. The $\beta$ here is the to-be-estimated parameter by the buyer. For simplicity, we assume the buyer initially has no knowledge about the model parameter $\beta$; formally, the buyer's initial belief $q = \mathcal{N}(0, I_d)$ about $\beta$ is a standard $d$-dimensional Gaussian.[5] The buyer would like to buy data from the seller to refine his Bayesian posterior belief about $\beta$, which is quite common in management sci-

ence (Liu et al., 2021b) and clinical trials (Berry, 2006). As a salient realistic feature of data pricing, we always assume the seller can only generate data of form $\widehat{y} = \langle \widehat{x}, \beta \rangle + \epsilon$ where $\epsilon \sim \mathcal{N}(0, \sigma(\widehat{x})^2)$ — that is, data are noisy responses for examined feature direction $\widehat{x}$. Moreover, as a natural statistical assumption, the larger the length $\|x\|$ of $x$ is, the larger the noise's magnitude is. Hence we assume the noise's variance scales with $\|x\|$, or formally, $\sigma(x) = \sigma(\frac{x}{\|x\|})\|x\|$ as widely adopted in statistical literature (Linnet, 1990; Kumar & Klefsjö, 1994; Bland & Altman, 1996; Wang & Mauldon, 2006).[6] Certainly, data are limited (which is why it has value). To capture its scarcity, we assume the seller is limited to producing at most $n$ data points of the above form, and the production has no cost.[7]

**Prior-free mechanism design and performance benchmark.** While $q$ is public knowledge, the decision context $x \in \mathbb{R}^d$ is the buyer's private information, also conventionally referred to as the buyer *type*, which is unknown to the seller. Unlike Bayesian mechanism design, we study *prior-free* mechanism design, which is known to be notoriously more challenging (Hartline et al., 2020; Hartline & Johnsen, 2021) but much more practical due to being free of assumptions on the seller's prior knowledge and on how buyer type is generated (Goldberg & Hartline, 2001; Devanur & Hartline, 2009). In this case, the performance benchmark is the "first-best" revenue, i.e., the revenue when perfectly knowing buyer type $x$ and producing $n$ data points based on $x$.

It turns out that solutions to this mechanism design question crucially hinge on the seller's power of customizing data production. We consider the following two natural settings.

- **Perfect Customization:** here, the seller has the flexibility to produce data for any direction $x$. Given such flexibility, the benchmark of the first-best is to produce $n$ responses for the buyer's type $x$ (assumed to be known in the benchmark) and then charge the buyer his value for this DPP. We denote this particular DPP as $g^n[x]$ which produces responses $y_i = \langle x, \beta \rangle + \epsilon_i$ for $i = 1, \cdots, n$. Hence, the benchmark revenue here is simply the buyer type $x$'s value for $g^n[x]$, which is $\mathtt{V}(g^n[x]; q, x)$.

- **Limited Customization:** here the seller has an existing set of data records $\{(x_i, y_i)\}_{i=1}^n$. Regardless of which buyer type $x$ is, the highest possible buyer value the seller can produce is to give all the data to the buyer. We denote this particular DPP as $g^X$, which is deter-

---

[5] Our results can be generalized to general Gaussian prior (same for every buyer), but with more complex notations.

[6] Otherwise, we only need to consider the length corresponding to the minimum relative noise in the direction, and this direction collapses to such a point.

[7] Another interpretation is that the data scarcity, i.e., at most $n$ data points, reflects the production cost.

mined by their data matrix $X$ and simply produces all their labels $\{y_i\}_{i=1}^n$. Hence, the benchmark revenue for buyer type $x$ is $\mathsf{V}(g^X; q, x)$.

**Convenient Notations.** If any mechanism achieves expected revenue that is *additively* within $\alpha$ of the benchmark described above for every buyer type $x$, we say this is an $\alpha$-regret mechanism.[8] A 0-regret mechanism is simply the mechanism that achieves the in-hindsight optimality. In this section, we always have $q = \mathcal{N}(0, I_d)$. For notational convenience, we will instead write the above value function as $\mathsf{V}(g; x)$ (only) in this section by ignoring $q$. Moreover, only the buyer knows his own type $x$, whereas the seller has no information at all about $x$. Hence, naturally, the buyer may *misreport* his type as some $\widehat{x}$, not necessarily equal to $x$, whenever this is more beneficial. We will always use $g[\widehat{x}]$ to denote the seller-designed DPP for a buyer report $\widehat{x}$. Like classic mechanism design, we say a mechanism is incentive-compatible (IC) if each buyer with type $x$ finds it optimal to report $x$ truly. A mechanism is individually rational (IR) if no buyer type has a negative expected utility.

### 4.1. The Power of Perfect Data Customization: Achieving the First Best

Our main result under perfect data customization is a surprisingly positive result showing that it is possible to have a 0-regret mechanism for selling DPPs that achieves the first-best in hindsight. Notably, such a full surplus charge has been shown to be impossible in classic item allocation problems (Myerson, 1981; Hartline et al., 2020; Hartline & Johnsen, 2021). This is fundamentally due to the different valuation functions in the two settings, as a function of allocated data. Our result demonstrates the surprising power of price discrimination brought by the special valuation of data as well as the flexibility of data customization.

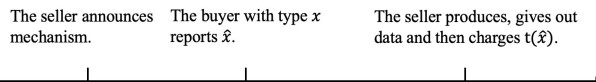

*Figure 1.* Timing line for perfect data customization.

**Theorem 4.1.** *With perfect data customization, the following Mechanism 1 with DPP $g[\widehat{x}] = g^n[\widehat{x}]$ and payment $t(\widehat{x}) = \frac{1}{2} \log(\frac{\sigma(\widehat{x})^2 + n}{\sigma(\widehat{x})^2})$ for any buyer reported type $\widehat{x}$ is IC, IR and 0-regret.*

Theorem 4.1 shows that the optimal allocation is to produce $n$ responses for type $x$. The key challenge is to find the optimal payment rule. As type $x$ is high-dimensional, traditional methods (Myerson, 1981) have become agnostic. We

---

[8]In online algorithm design, additive loss from the optimal in hindsight is often denoted as "regret" whereas multiplicative loss is called "competitive ratio".

---

**Algorithm 1** An Optimal Mechanism under Perfect Data Customization

**Input:** buyer's report $\widehat{x}$
Seller charges buyer $t(\widehat{x}) = \frac{1}{2} \log(\frac{\sigma(\widehat{x})^2 + n}{\sigma(\widehat{x})^2})$
Seller produces $n$ responses for the same $\widehat{x}$: $\widehat{y}_i = \langle \widehat{x}, \beta \rangle + \epsilon_i$, and reveals $\{\widehat{y}_i\}_{i=1}^n$ to the buyer.

---

conclude that the payment rule for allocation $g[\cdot]$ must have the following form that $t(x) = \int_{x_0}^{x} \nabla_y \mathsf{V}(g[y]; s)\big|_{y=s} \cdot ds + t(x_0)$.

Compared with the well-known results of single-dimension in Myerson (1981), there are two main differences considering a multi-dimensional type. First of all, we need to generalize the derivation of a one-dimensional function to the gradient computation in the case of higher dimensions. Second, when calculating the payment rule $t(\cdot)$, the integral result is required to be independent of the initial starting point and the integral path we choose. In our situation, it means that no matter what $x_0$ and the path from it to the final $x$ we choose, the result of the integral should be the same. These conditions constrain the application scenarios of the extended method for multi-dimensional mechanism design, as we only need monotonicity for the single-dimensional situation. Nevertheless, the proof will elaborate that our definition of instrumental value satisfies all these conditions endogenously, highlighting the importance of our framework of instrumental value and suggesting its broad range of applications.

### 4.2. Selling Existing Data Records with Limited Customization

We now turn to the situation when the seller has $n$ data points $\{(x_i, y_i)\}_{i=1}^n$ at hand and can only process this existing dataset and sell it to the buyer, where $\Sigma(X) = \text{diag}\{\sigma(x_1)^2, ..., \sigma(x_n)^2\}$ and $Y = (y_1, ..., y_n)^T$. The design matrix $X$ is public knowledge, but the buyer does not observe the corresponding responses $Y$. In practice, for example, in clinical trials, information about the subject group is usually known, but the evaluation of the drug's effectiveness needs to be purchased. Sellers in real-world scenarios usually disclose data quality. We model it by $\Sigma(X)$, which represents the measurement error or the specifications of the experimental instruments, such as their precision. We first note that while our benchmark here is the buyer's maximum possible value $\mathsf{V}(g^X; x)$, this benchmark is impossible to obtain due to not knowing the buyer's type, since if the seller indeed reveals all these data records to every buyer, then every buyer type would want to misreport his type to be some $\widehat{x}$ that has the smallest $\mathsf{V}(g^X; \widehat{x})$. Hence, some data processing based on $\{(x_i, y_i)\}_{i=1}^n$ is necessary to achieve price discrimination and get the mechanism close to the

benchmark $V(g^X; x)$ for *every* $x$. Indeed, it turns out that in this case, perhaps as expected, there is no 0-regret mechanism. However, surprisingly, we nevertheless show that there is an IC and IR mechanism with small constant regret.

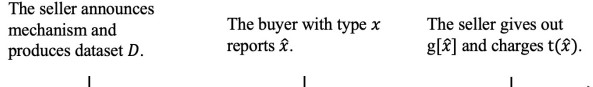

The seller announces mechanism and produces dataset $D$.

The buyer with type $x$ reports $\widehat{x}$.

The seller gives out $g[\widehat{x}]$ and charges $t(\widehat{x})$.

*Figure 2.* Timing line for limited data customization.

**Theorem 4.2.** *When the type space is continuous, there is no 0-regret mechanism under limited customization.*

**Theorem 4.3.** *The regret of the following SVD mechanism under limited data customization is at most* $\log(\kappa((\sqrt{\Sigma(X)})^{-1}X))$, *where* $\kappa((\sqrt{\Sigma(X)})^{-1}X)$ *is the square root of the ratio between the largest and the smallest singular values of* $X^T \Sigma(X)^{-1} X$.

---

**Algorithm 2** The SVD Mechanism

**Input:** buyer's report $\widehat{x}$, seller's design matrix $X$

Normalize $(X, Y) \leftarrow ((\sqrt{\Sigma(X)})^{-1}X, (\sqrt{\Sigma(X)})^{-1}Y)$ and announce normalization

Perform SVD over $X$, obtaining $X = U \begin{bmatrix} S \\ 0_{(n-d) \times d} \end{bmatrix} V$

where $S = \mathrm{diag}\{\lambda_1, \ldots, \lambda_d\}$

Let $L$ denote the left inverse of $X$ where $L = U \begin{bmatrix} S^{-1} \\ 0 \end{bmatrix} V$

Define the mapping $b(\cdot) : \mathbb{R}^d \to \mathbb{R}^n$ where $b(\widehat{x}) = \frac{L\widehat{x}}{\|L\widehat{x}\|}$

**Output:** $g[\widehat{x}] = (b(\widehat{x})^T X, b(\widehat{x})^T Y)$, $t(\widehat{x}) = \frac{1}{2}\log(\widehat{x}^T\widehat{x}) - \frac{1}{2}\log(\widehat{x}^T(I + X^T b(\widehat{x})b(\widehat{x})^T X)^{-1}\widehat{x})$.

---

Theorem 4.3 gives an illustration of the ability of price discrimination under limited customization. Unlike perfect customization, the seller cannot achieve the first-best revenue in all cases. In other words, the worst-case distance between optimal revenue and the first-best revenue is strictly positive, which implies that limited customization limits the level of market unfairness. However, even though the buyer could get a non-zero consumer surplus, it is still small, and he still suffers inequality against the seller.

The following results further show that for certain $X$, the seller can achieve the first-best revenue and 0-regret, highlighting the potential unfairness in the data market even under limited customization.

**Corollary 4.4.** *When the seller has isotropic data, i.e., all singular values of* $X^T\Sigma(X)^{-1}X$ *are the same, there exists a mechanism that satisfies both IC and IR, and achieves 0-regret.*

At the same time, if the private types of the buyer, which are public knowledge, have certain structures, it's also possible for the seller to achieve the first-best revenue. For example,

we find a new information structure called *multi-armed bandits setting* (cf. Appendix D.2) which also leads to 0-regret for the seller. Therefore, we know that whether 0-regret is achievable depends on the information leakage. In Corollary 4.4, since every direction confers equal value, it discloses the buyer's willingness to pay. In the MAB setting, the intuition is that other directions provide no information, discouraging the buyer from submitting untruthful reports and naturally leading to price discrimination.

## 5. Conclusion and Discussion

Compared to a dataset's intrinsic value, the instrumental value better captures its economic value in the market. This paper first introduces a principled way to quantify a dataset's instrumental value, and then studies how to apply this valuation to a data pricing problem. The designed mechanism illustrates the surprising power of data customization, which may help data sellers to do first-degree price discrimination, leaving zero surplus to buyers. Even under limited data customization, the flexibility of selling any derivatives or statistics from an existing dataset already gives the seller significant power to discriminate against buyers. These results hint at a potential need for regulations in order to foster a sustainable data market. On the technical side, our data valuation's connection to the value of information (Frankel & Kamenica, 2019) may be of independent interest. Moreover, our designed mechanisms for multi-dimensional buyer preference, particularly the mechanism based on the singular value decomposition, may be of both theoretical and practical interest.

Several open questions are worth future investigation. Is it feasible to explicitly resolve the mechanism design challenge under dissimilar information flows? Our data valuations are derived from linear regression; how transferable are they to data valuations under other learning models? Previous works have shown that Data Shapley often has a similar order when computed using different ML models (Jia et al., 2021; Schoch et al., 2022). Moreover, given that we have demonstrated the potentially high-skewed surplus allocation between the seller and buyer, how should regulatory authorities step in (Mas-Colell et al., 1995; Jehle, 2001) to sustain market efficiency? What would be the implications of multiple sellers operating in the market, or if it were a perfectly competitive market? We leave these intriguing follow-up questions as potential avenues for future research.

## Acknowledgements

Zhaoran Wang acknowledges National Science Foundation (Awards 2048075, 2008827, 2015568, 1934931), Simons Institute (Theory of Reinforcement Learning), Amazon, J.P. Morgan, and Two Sigma for their supports. Zhuoran Yang acknowledges support from NSF DMS 2413243. Haifeng Xu is supported by the AI2050 program at Schmidt Sciences (Grant G-24-66104), Army Research Office Award W911NF23-1-0030, and NSF Award CCF-2303372. The authors would like to thank the reviewers for their valuable comments and suggestions, which have greatly improved the article.

## Impact Statement

This paper presents work whose goal is to advance the field of Machine Learning. There are many potential societal consequences of our work, none which we feel must be specifically highlighted here.

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

# Appendix for "An Instrumental Value for Data Production and its Application to Data Pricing"

## A. Omitted Details in Section 1

### A.1. Related Works

**Valuating data and information:** The paper is related to existing works for valuating data and information. The existing literature has been extensively focused on Shapley value (or variants thereof) for valuing data (Schoch et al., 2022; Ghorbani & Zou, 2019; Jia et al., 2019), which is a principled way to "fairly" attribute (Roth, 1988) the contribution of each data point/set to an ML task. This is why the Shapley value is an average contribution, averaged over all other possible training situations. Later works also realize that equal-weight average may not be ideal for ML tasks, hence proposed variants of Data Shapley that prioritize some training situation in the average (e.g., beta-Shapley (Kwon & Zou, 2021)). Recently, Guo et al. (2025) considers how to value data in human-AI decision-making. In contrast, our data value is instrumental and is to quantify a dataset's *marginal* contribution. Hence, the meaning of this value is not to average over many possible training situations but instead to look at a marginal increase over the previous best training situation. Contemporaneously, Choe et al. (2024) uses a gradient-based data valuation for large language models, and we stem from Bayesian regression.

**Selling data:** For existing works on selling data, Segura-Rodriguez (2021) studies selling data to a buyer who wishes to minimize the quadratic loss of an estimate. The biggest difference that separates our work from this work is the buyer's private type. In our case, we assume the features of the observations that the buyer wishes to obtain labels for are unknown to the seller. On the other hand, Segura-Rodriguez (2021) assumes the buyer's features are known a priori, but the type of label the buyer wishes to estimate is the unknown private type. For instance, if a production company purchases data from Nielsen, our work corresponds to the setting that the TV show the company is producing is its private type and is unknown to Nielsen, whereas Segura-Rodriguez (2021) assumes the TV show itself is known, but Nielsen does not know if the production company wants to predict the show's ratings or the show's audience demographics. The difference in the information structure leads to fundamental differences between our work and theirs, both in terms of the proposed mechanisms and in terms of the theoretical analyses. Chen et al. (2022) studies the problem of selling data to a machine learner, using a costly signaling step to ensure that the buyer can accurately evaluate the quality of the data purchased. Our work eschews the step but leverages the properties of instrumental value in the Bayesian linear regression setting. Chen et al. (2019) values the data based on the model that the buyer learns from the data. Agarwal et al. (2019) studies a two-sided data market with multiple data buyers and sellers, in which the buyers compete with one another by bidding for data. The latter two works, however, assume that the buyer's learned model is observable by the seller, which avoids the complication introduced by the private type considered in our work.

**Selling information:** Our work is also related to the long line of work on selling information (Babaioff et al., 2012; Hörner & Skrzypacz, 2016; Admati & Pfleiderer, 1988; Kastl et al., 2018; Xiang & Sarvary, 2013; Bergemann & Bonatti, 2015; Liu et al., 2021a) and we refer interested readers to Bergemann & Bonatti (2019) for a survey of these works. Of these works, Babaioff et al. (2012) studies the optimal one-round revelation mechanisms for selling signals, assuming that the buyer's type space is discrete and finite, and Chen et al. (2020) extends the problem setting to one where the buyer also has finite budget. Bergemann et al. (2018) and later work Bergemann et al. (2022) focuses on the optimal sales of Blackwell experiments under the assumption that the buyer's type space is finite. Hartline et al. (2020); Hartline & Johnsen (2021) study optimal type-prior independent approximation multiplicative factor, whereas we focus on the corresponding addictive factor and pave the way for prior-independent mechanism design in an addictive manner.

There is also a rich body of literature on active learning (Freund et al., 1997; Settles, 2009), studying the effect of data sequence. However, active learning studies how to select sequential data and train models to maximize accuracy, whereas our paper studies how to define a valid data valuation function that captures downstream users' utilities. We approach this study from a utilitarian perspective, and our value corresponds to information gain only in a very canonical special case.

## B. Omitted Details in Section 2

### B.1. Omitted Details in Section 2.1

We now give omitted algebra details of Example 1.

*Example* 4 (Examples of DPPs). Consider a linear model $y = \langle x, \beta \rangle + \epsilon$. A learner would like purchase data to predict $y = \langle x, \beta \rangle$ for $x = (1, 1)$. A simple example of DPPs — described by two feature directions $x_1 = (1, 3), x_2 = (3, 1)$ — simply produces response variables corresponding to the given $x_1, x_2$: $y_1 = \langle x_1, \beta \rangle + \epsilon_1 = -1$ and $y_2 = \langle x_2, \beta \rangle + \epsilon_2 = 1$.

In this case, $D = \{y_1, y_2\}$ is the realized data which is produced by data production process $g_\beta(D) = \mathbb{P}_\epsilon(y_1 - \langle x_1, \beta\rangle) \cdot \mathbb{P}_\epsilon(y_2 - \langle x_2, \beta\rangle)$. Hence the DPP here is fully described by two feature directions $x_1, x_2$. The realized data $D = \{y_1, y_2\}$, together with the knowledge of DPP $g$, can help the learner to refine the estimation of $\beta$.

Instead of directly producing data $y_1, y_2$ for directions $x_1, x_2$, one could also produce the data $D = \bar{y} = \frac{y_1 + y_2}{2}$ for feature direction $\bar{x} = \frac{x_1 + x_2}{2}$ using DPP with $g_\beta(\bar{y}) = \mathbb{P}_{\epsilon_1, \epsilon_2}(\bar{y} - \langle \bar{x}, \beta\rangle)$ where the probability density is over the randomness of both noise terms $\epsilon_1, \epsilon_2$. Notably, this DPP is subtly different from directly generating the response $y = \langle \bar{x}, \beta\rangle + \epsilon$ because $\bar{y}$ has a smaller variance than $y$, despite having the same mean, hence is strictly more informative. Indeed, to produce data $D = y = \langle \bar{x}, \beta\rangle + \epsilon$, the data owner would need to add extra noise to $\bar{y} = \frac{y_1 + y_2}{2}$ in order to match its distribution. This illustrates the rich space of DPPs.

Perhaps more interestingly, we could also design a DPP that produces data $y^\perp = y_1 - y_2 = -2$ for direction $x^\perp = x_1 - x_2 = (-2, 2)$. This DPP is of no interest to the buyer above since $\bar{x} \perp x$, but nevertheless its realized data $D = y^\perp$ carries information about $\beta$ and hence may be of interest to other buyers.

Suppose the learner has an uninformative Gaussian prior $\beta \sim \mathcal{N}(0, I)$ and the noise in response variable follows the standard Gaussian $\mathcal{N}(0, 1)$. Then the data generating function $g_\beta(y)$ for a $DPP(x)$ that produces data $y = \langle x, \beta\rangle + \epsilon$ is $g_\beta(y) = \frac{1}{\sqrt{2\pi}} \exp(-(y - \langle x, \beta\rangle)^2/2)$. If we produce data $\bar{y} = \frac{y_1 + y_2}{2}$ as the average response variable for $x_1, x_2$, the posterior of $\beta$ prescribed in Equation (2) can be calculated as $\mathcal{N}(0, \begin{bmatrix} 9/17 & -8/17 \\ -8/17 & 9/17 \end{bmatrix})$ and the variance of the to-be-predicted variable $\langle x, \beta\rangle$ reduces from 2 to $\frac{2}{17}$, same as fully revealing original data records $\{(x_1, y_1), (x_2, y_2)\}$. However, if we only reveal $(x^\perp, y^\perp)$, the posterior is $\mathcal{N}((2/5, -2/5), \begin{bmatrix} 3/5 & 2/5 \\ 2/5 & 3/5 \end{bmatrix})$ and the variance remains 2. Notably, these improvements can all be calculated without knowing the realized data. This will be useful for selling DPPs since it means the value of a DPP can be quantified without seeing the realized data. This resolves the concern that data will lose its value after being seen.

We here give a rigorous definition of concave functionals and generalized Bregman divergence in Definition 2.2.

**Definition B.1.** Let $\mathcal{B}$ denote the domain of parameter $\beta$ and consider *functional* $F : \Delta(\mathcal{B}) \to \mathbb{R}$ that maps the space of probability distributions over $\mathcal{B}$ to a real value. We say *functional* $F(\cdot)$ is concave if for any pair of distributions $p, q \in \Delta(\mathcal{B})$ and any $\lambda \in [0, 1]$, we have
$$F(\lambda p + (1 - \lambda)q) \geq \lambda F(p) + (1 - \lambda)F(q).$$
Moreover, we say $\nabla F(\cdot)$ is its (functional) superdifferential if for any $p, q \in \Delta(\mathcal{B})$ we have
$$F(q) + \langle \nabla F(q), p - q\rangle \geq F(p).$$
The generalized Bregman divergence induced by functional $F$ is defined as
$$D_F(p, q) = F(q) - F(p) + \langle \nabla F(q), p - q\rangle.$$

## B.2. Omitted Details in Section 2.2

Next, we are ready to give more details on the characteristics of $\mathtt{val}(\cdot; \cdot, \cdot)$ and $\mathtt{cost}(\cdot, \cdot)$.

**Definition B.2 (Valid Cost of Uncertainty (Definition 2.5 Continued)).** Relatedly, the corresponding $\mathtt{cost}(q, x) = \mathtt{val}(\vee; q, x)$ is called a valid *cost of uncertainty* function. Moreover, in this case, we say the $\mathtt{val}(\vee; q, x)$ and $\mathtt{cost}(q, x)$ are *coupled* (since they correspond to the same CBDM).

**Proposition B.3 (Characterization of Valid Valuation Functions).** *A valuation function* $\mathtt{val}$ *is valid if and only if it satisfies the following properties simultaneously*

1. ***No value for null data:*** $\mathtt{val}(\emptyset; q, x) = 0$ *for any prior* $q$;

2. ***Positivity:*** $\mathtt{val}(D; q, x) \geq 0$ *for any realized data* $D$ *and any prior* $q$ *over parameter* $\beta$;

3. ***Invariance to data acquisition orders:*** *The expected value of data is invariant to the order of data acquisition. Formally, given any prior* $q$ *over* $\beta$ *and any two data production process* $g_1$ *and* $g_2$, *let* $p_1 = p(\cdot|q, D_1)$ *and* $p_2 = p(\cdot|q, D_2)$ *be the posteriors updated by realized data* $D_1$ *and* $D_2$ *respectively. Then, we have*
$$\mathbb{E}_{D_1, D_2}[\mathtt{val}(D_1; q, x) + \mathtt{val}(D_2; p_1, x)] = \mathbb{E}_{D_1, D_2}[\mathtt{val}(D_2; q, x) + \mathtt{val}(D_1; p_2, x)].$$

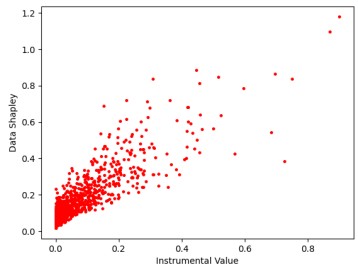 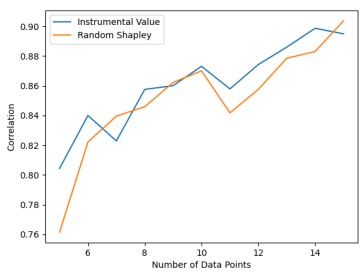 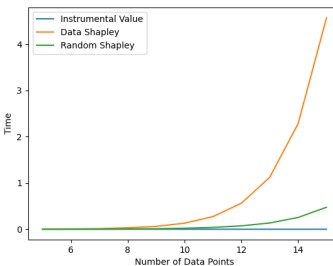

(a) Scatter plot between instrumental value and Data Shapley.

(b) Correlation of instrumental value and Random Shapley with Data Shapley.

(c) Computation time of instrumental value, Data Shapley and Random Shapley.

*Figure 3.* Comparison between instrumental value, Data Shapley and Random Shapley.

The first two properties are both natural. The third property indicates that the order of obtaining data from two data production processes should not change the total expected value of these two DPP. This is also a natural property for any valid valuation function. Interestingly, it turns out that these three properties suffice to guarantee the validity of `val` in the sense of Definition 2.5 and Equation (3).

The following proposition offers a characterization for the validity of a cost of uncertainty function $\mathtt{cost}(q, x)$.

**Proposition B.4.** *A cost of uncertainty function* $\mathtt{cost}(q, x)$ *is valid if and only if it satisfies the following properties simultaneously*

1. ***0 cost under certainty:*** *when the prior $q$ degenerates to a Dirac delta distribution over $\beta$, the cost of uncertainty is $0$, i.e.,* $\mathtt{cost}(\delta_\beta, x) = 0$ *for any $x$.*

2. ***Concavity:*** $\mathtt{cost}(q, x)$ *is concave in $q$ for any $x$.*

The first property is obvious from the definition. To see the second property intuitively, let $q = \lambda q_1 + (1 - \lambda)q_2$ for some $\lambda \in [0, 1]$ and beliefs $q_1, q_2$, and consider the following situation. Suppose there is some data informing the buyer that $\beta$ is drawn either from distribution $q_1$ (with probability $\lambda$) or from $q_2$ (with probability $1 - \lambda$). If the buyer must make a decision prior to observing the signal, his cost of uncertainty is $\mathtt{cost}(\lambda q_1 + (1 - \lambda)q_2)$. If the buyer is allowed to make a decision after observing the signal, his expected cost of uncertainty would be $\lambda\mathtt{cost}(q_1) + (1 - \lambda)\mathtt{cost}(q_2)$, which by concavity is no greater than that for when he is not allowed to observe the signal. Thus, concavity is a natural formalization of the intuition that having more information on the parameter distribution will not increase the cost of uncertainty (in this case, the additional information is the signal pinning down whether $\beta$ is drawn from $q_1$ or $q_2$). What is surprising, however, is that these two natural properties suffice to pin down any valid functions for the cost of uncertainty.

### B.3. Omitted Details in Section 2.3

B.3.1. NUMERICAL STUDY

We conducted numerical experiments using Python 3.9.15 on a server equipped with dual Intel(R) Xeon(R) Platinum 8260 CPUs. We use entropy reduction as an illustrative example of $\mathtt{val}(\cdot; \cdot, \cdot)$. We first show that the instrumental value has a high correlation with Data Shapley, hinting at consistent underlying economic intuition between them. Since the time complexity of computing Data Shapley is exponential to the size of the dataset, we choose $n = 10$ and the dimension $d = 5$. We randomly generate $D$ with some mean and variance. Then, we randomly sample $x$ of interest with some other mean and variance. We assume $q = \mathcal{N}(0, 1)$. We sample 1000 trials in parallel and observe in Figure 3(a) that Data Shapley is generally larger (around twice) than the instrumental value which agrees with our theory. In the meanwhile, they share a high correlation, say 0.8650, implying consistent economic intuition. This phenomenon inspires us to study whether we can use instrumental value to approximate Data Shapley. When we calculate the empirical Data Shapley using the mean of 102 samples, we call it Random Shapley. In this case, we obtain almost the same correlation of 0.8645. However, the average computing time is 0.1380, 0.0002 and 0.0218 for Data Shapley, instrumental value and Random Shapley

respectively, showing the great potential of instrumental value for computational traceability and offering a new perspective for calculating Data Shapley.

Next, we empirically study the time complexity of instrumental value, Data Shapley and Random Shapley, as well as the promising potential of using instrumental value to approximate Data Shapley instead of Random Shapley. We now range size $n$ from 5 to 15. For Random Shapley, we set the sample size to $\frac{2^n}{n}$ and it has a similar correlation with Data Shapley as instrumental value as shown in Figure 3(b). Nonetheless, the instrumental value has a great advantage in computation time. We detail the average computing time in Table 1. It enjoys polynomial time complexity with respect to size $n$ while Data Shapley and Random Shapley suffer from exponential curse. See Figure 3(c) for reference. Therefore, instrumental value has promising application prospects in the industrial sector. It is also of independent interest to implement our valuation on real-world data platforms.

| Size | Instrumental Value | Data Shapley | Random Shapley |
|------|--------------------|--------------|----------------|
| 5  | 0.000129 | 0.003475 | 0.001134 |
| 6  | 0.000130 | 0.007387 | 0.002147 |
| 7  | 0.000116 | 0.012579 | 0.003002 |
| 8  | 0.000180 | 0.030939 | 0.006260 |
| 9  | 0.000202 | 0.060909 | 0.010930 |
| 10 | 0.000202 | 0.130378 | 0.020986 |
| 11 | 0.000195 | 0.273053 | 0.039111 |
| 12 | 0.000215 | 0.560476 | 0.072644 |
| 13 | 0.000240 | 1.120840 | 0.134825 |
| 14 | 0.000227 | 2.280680 | 0.253490 |
| 15 | 0.000239 | 4.566204 | 0.474172 |

*Table 1.* Average computing time among three valuations.

## C. Omitted Details in Section 3

We would like to give the concrete definition of "entropy" here.

**Definition C.1** (Differential Entropy (Thomas & Joy, 2006))**.** The differential entropy $h(Z)$ of a continuous random variable $Z$ with density $\mu(z)$ is $h(Z) = -\int_{z \in S} \mu(z) \log(\mu(z)) dz$, where $S$ is the support of the random variable $Z$.

## D. Omitted Details in Section 4

### D.1. Omitted Details in Section 4.1

For simplicity, we use $\mathtt{V}(x, \widehat{x})$ to represent $\mathtt{V}(g[\widehat{x}]; x)$ when easy to infer from context. Let's consider a wider range of DPPs that the seller can add man-made noise $\epsilon' \sim \mathcal{N}(0, \delta(x)^2)$ when revealing $y$. That is to say, the seller will reveal $(x, y + \epsilon')$ rather than $(x, y)$. After that, we are going to show that adding no noise, i.e., $\delta(x) = 0$, can lead to 0-regret and it's definitely what the seller will choose.

Concerning the definition of $\mathtt{V}(\cdot; \cdot, \cdot)$ in Equation (4), we can establish a lemma exhibiting the concrete instrumental value of data in detail.

**Lemma D.1.** *If the seller gives the buyer $n$ data points, it holds that the instrumental value of data for the buyer is*

$$\mathtt{V}(x, \widehat{x}) = \mathtt{V}(g^n[\widehat{x}]; \mathcal{N}(0, I_d), x) = \frac{1}{2} \log\left(x^T x\right) - \frac{1}{2} \log\left(x^T \left(I + \frac{n \widehat{x} \widehat{x}^T}{\sigma(\widehat{x})^2 + \delta(\widehat{x})^2}\right)^{-1} x\right).$$

By revelation principle (Gibbard, 1973; Holmström, 1978; Myerson, 1979), we know that any social choice function implemented by an arbitrary mechanism can be implemented by an incentive-compatible-direct-mechanism with the same equilibrium outcome. Therefore, we focus on mechanisms that satisfy incentive-compatibility (IC) constraint and at least one optimal mechanism belongs to this subclass. In the context of our model, incentive-compatibility constraint means that the buyer has the motivation to report his own private type truthfully considering the allocation rule $g[\cdot]$ and payment rule

$t(\cdot)$. To ensure the buyer is motivated to purchase data, we further impose individual rationality (IR) constraints. It means that a rational buyer only participates in a transaction when his expected revenue is no less than zero. Formally, the IC and IR constraints for a mechanism $(g[\cdot], t(\cdot))$ are

IR
$$\mathtt{V}(x,x) \geq t(x) \text{ for all } x,$$

IC
$$\mathtt{V}(x,x) - t(x) \geq \mathtt{V}(x,\widehat{x}) - t(\widehat{x}) \text{ for all } x, \widehat{x}.$$

In other words, IC constraint means that truthful reporting is a dominant strategy for the buyer and he has no incentive to report a wrong type as there are no benefits by doing so. IR constraint gives the buyer a reason to join the market and motivates the transaction as his expected utility is non-negative.

Since we assume $\sigma(x) = \sigma(\frac{x}{\|x\|})\|x\|$, it is evident that we could solely consider the direction of the type, rather than its length. This can be achieved by substituting $x$ and $\widehat{x}$ with $\frac{x}{\|x\|}$ and $\frac{\widehat{x}}{\|\widehat{x}\|}$, respectively. We then have

$$\mathtt{V}(x, \widehat{x}) = \mathtt{V}(\frac{x}{\|x\|}, \frac{\widehat{x}}{\|\widehat{x}\|}).$$

We thus restrict our attention to types with a unit norm, i.e. $\mathcal{S}^{d-1} = \{x : \|x\| = 1\}$, which is equivalent to considering a general space in $\mathbb{R}^{d-1}$. Inspired by the method in Myerson (1981), we have the following lemma when $x$ has multiple dimensions.

**Lemma D.2.** *For any payment rule satisfying the IC constraint, it holds that*

$$\nabla t(x) = \nabla_y \mathtt{V}(x,y)\big|_{y=x}.$$

*Therefore, it leads to the following form of payment rule that*

$$t(x) = \int_{x_0}^{x} \nabla_y \mathtt{V}(s,y)\big|_{y=s} \cdot ds + t(x_0) = \frac{1}{2} \log(\frac{\sigma(x)^2 + \delta(x)^2 + n}{\sigma(x)^2 + \delta(x)^2}) + C,$$

*and it's independent of the selection of $x_0$ and the path of integration.*

Therefore, for any $\delta(x)$, we have the corresponding optimization problem for optimal mechanism, that is

$$\max_{C, \delta(\cdot)} \mathbb{E}_x t(x) = \mathbb{E}_x [C + \frac{1}{2} \log(\frac{\sigma(x)^2 + \delta(x)^2 + n}{\sigma(x)^2 + \delta(x)^2})] \mathbb{1}(\text{IR holds}),$$

where

$$\mathbb{1}(\text{IR holds}) = \mathbb{1}(C + \frac{1}{2} \log(\frac{\sigma(x)^2 + \delta(x)^2 + n}{\sigma(x)^2 + \delta(x)^2}) + \frac{1}{2} \log(x^T (I + \frac{nxx^T}{\sigma(x)^2 + \delta(x)^2})^{-1} x) \leq 0).$$

Though information gain is a widely used metric, our analysis method can be extended to a wider family of instrumental values as long as keeping the concavity property as what $\log(\cdot)$ does. We add the following remark, inspired by $f$-divergence (Rényi, 1961), to show how to generalize our results.

*Remark* D.3. The method can be easily extended to any $\mathtt{V}(\cdot, \cdot)$ function following the form

$$\mathtt{V}(x, \widehat{x}) = f(\frac{x^T x}{x^T (I + \frac{\widehat{x}\widehat{x}^T}{(\sigma(\widehat{x})^2 + \delta(\widehat{x})^2)/n})^{-1} x}),$$

where $f(\cdot)$ is a concave function, and the results remain valid when $t(x)$ is of the form

$$t(x) = C + f(\frac{\sigma(x)^2 + \delta(x)^2 + n}{\sigma(x)^2 + \delta(x)^2}).$$

With the characteristics of $t(\cdot)$, we can conclude that when the seller is embedded with the ability of perfect customization, she can obtain the first-best revenue and leave zero consumer surplus to the buyer, say Theorem 4.1. It reflects the price discrimination scenario in the data market and shows corresponding unfairness which is quite different than the situation in other markets.

## D.2. Omitted Details in Section 4.2

Let us recall what data the seller has. The seller produces $n$ data points $\{(x_i, y_i)\}_{i=1}^n$ consisting dataset (DPP resp.) $D$ ($g^X$ resp.). Here, $y_i = x_i^T \beta + \epsilon$, where $\epsilon$ follows a normal distribution with mean 0 and variance $\sigma(x_i)^2$. For simplicity, let $X \in \mathbb{R}_{n \times d}$ denote the design matrix of the whole dataset and $Y$ the corresponding responses. We use $\Sigma(X)$ to represent the noise covariance matrix, i.e. $\Sigma(X) = \text{diag}\{\sigma(x_1)^2, ..., \sigma(x_n)^2\}$.

In this particular type of information flow, the buyer will not experience as much unfairness as the one in the situation described in Section 4.1. This implies that offering customization options may have negative consequences for society, and raises questions about how to regulate market power in the data market. However, it is important to note that the buyer's surplus will be bounded from above by a constant unrelated to the buyer's type $x$, which means that in a monopolized market, buyers will likely find it difficult to attain high surplus.

Similar to Section 4.1, we only consider mechanisms that satisfy both IC and IR constraints, that is,

IR $\qquad\qquad\qquad\qquad V(x, x) \geq t(x)$ for all $x$,

IC $\qquad\qquad\qquad\qquad V(x, x) - t(x) \geq V(x, \widehat{x}) - t(\widehat{x})$ for all $x, \widehat{x}$.

As we discussed before, if it's impossible to achieve the first-best revenue, i.e., 0-regret, in this subclass of mechanisms, there won't exist a mechanism achieving the first-best revenue among all unconstrained mechanisms.

However, as demonstrated by Theorem 4.3, even though the seller cannot achieve the first-best revenue, the buyer's surplus is also upper bounded by a constant relative only to the pre-produced $X$, rather than his personal type $x$. It is important to note that the buyer may have varying consumer surpluses for different personal types, while the union of upper bounds holds for all types $x$. The proof of Theorem 4.3 leads directly to the following corollary about the seller that she can achieve within an additive factor of $\log(\kappa)$ of the first-best revenue.

**Corollary D.4.** *For any pre-produced dataset $X$, there exists a mechanism in which the expectation of the seller's revenue is at most $\log(\kappa((\sqrt{\Sigma(X)})^{-1}X))$ less than the first-best revenue.*

Now, let's give the definition of multi-armed bandits setting in our data pricing scenario.

**Definition D.5.** We call a setting having a form of multi-armed bandits if every type vector $x$ is a member of the standard basis of a $d$-dimensional Euclidean space $\mathbb{R}^d$. In other words, only one component of $x$ is one and the others are all zeros.

It is closely related to multi-armed bandits, the famous machine learning model. In multi-armed bandits, observations of rewards of one arm contain no information about rewards of other arms. In the data pricing, the multi-armed bandits setting means that data points associated with type $x$ involve no information about other types. Using a geometric perspective, it illustrates that the information corresponding to different types is orthogonal. Now, we have a corollary in which the seller can obtain the first-best revenue.

**Corollary D.6.** *In the multi-armed bandits (MAB) setting, there exists a mechanism that satisfies both IC and IR, and achieves the first-best revenue, i.e., 0-regret.*

Together with Theorem 4.3, Corollaries 4.4 and D.6 show that the quantity of the consumer surplus not only depends on the concrete properties of the dataset $X$, but probably related information contained in the structure of the buyer's private types as well. In light of the universal Theorem 4.3, insofar as the diverse directions of data hold distinct values for the buyer, the seller harbors a motivation to induce the buyer, possessing a greater willingness to pay, to disclose his authentic type. To devise a truthful mechanism, the seller must relinquish some charge in specific instances to attain the maximum expected revenue. Notwithstanding, as per Corollary 4.4, if each direction is isotropic, then it will confer equal value to the buyer. Hence, it discloses the buyer's willingness to pay, which is pivotal for the seller to implement price discrimination. It illuminates the crucial point of mechanism design in the data market, where the seller's focus is not on the buyers' individual type, but rather on their willingness to pay. Corollary D.6 however highlights the role of information leakage. In the context of multi-armed bandits, data from other arms provides no information then no benefit to the buyer, dissuading the buyer from untruthful reports, creating a natural case of price discrimination.

# E. Omitted Proof in Section 2

The proofs for Proposition 2.6, Proposition B.4, and Theorem 2.7 are closely related.

For the only if direction, i.e. showing that the desiderata imply the existence of some corresponding decision-making problem, all three results share the same underlying constructed instance. As such, we organize our proof by first introducing a series of propositions and lemmas that will be essential for the proof, and then justifying the results by combining these results.

We introduce two concepts, measure of information and measure of uncertainty, which serve as the statistical analog to Definition 2.5 and are crucial to formally establishing the results in Section 2.

**Definition E.1.** Any functional $d(\mu, \nu) : \Delta(\mathcal{Y}) \times \Delta(\mathcal{Y}) \to \mathbb{R}$ is a **measure of information** and any $H(\mu) : \Delta(\mathcal{Y}) \to \mathbb{R}$ is a **measure of uncertainty**. We say $d$ and $H$ are **coupled** if for any signaling scheme $s$, we have $\mathbb{E}[d(\mu(s), \mu)] = \mathbb{E}[H(\mu) - H(\mu(s))]$, where $\mu(s)$ denotes the posterior distribution given by Bayes rule after observing the signal $s$.

The conditions imposed in Propositions 2.6 and B.4 can then be similarly defined as follows.

**Definition E.2.** We say a measure of information $d$ satisfies **no value for the null data** dubbed **null-information** if $d(\mu, \mu) = 0$ for all $\mu \in \Delta(\mathcal{Y})$, **positivity** if $d(\mu, \nu) \geq 0$ for all $\mu, \nu \in \Delta(\mathcal{Y})$, and **invariance to data acquisition** dubbed **order-invariance** if for any pair of signaling schemes $\pi_{s_1}$ and $\pi_{s_2}$, the expected information gain is independent of the order in which the signals $s_1$ and $s_2$ are observed, i.e.

$$\mathbb{E}_{s_1, s_2}[d(\mu(s_1), \mu) + d(\mu(s_1 \cap s_2), \mu(s_1))] = \mathbb{E}_{s_1, s_2}[d(\mu(s_2), \mu) + d(\mu(s_2 \cap s_1), \mu(s_2))].$$

**Definition E.3.** We say a measure of uncertainty $H$ satisfies **0 cost under certainty** dubbed **null-uncertainty** if $H(g_\beta) = 0$ for any $\beta$ and **concavity** if $H$ is a concave functional over $\Delta(\mathcal{Y})$.

For the rest of the proof, we heavily make use of the notions $d$ and $H$, for they directly measure the distance between distributions or the uncertainty within some distribution. Compared with the economic formulation in Section 2, these notions are easier to manipulate mathematically. We begin by first showing $d$ should be the Bregman divergence of $H$.

We use $\mu, \nu$ in Lemma E.4 to emphasize that the result holds for generic distributions $\mu, \nu \in \Delta(\mathcal{Y})$, not necessarily those induced by some belief over the parameter space and a suitable model $g$.

**Lemma E.4.** *Let $d(\mu, \nu)$ denote some measure of distance between distributions $\mu, \nu \in \Delta(\mathcal{Y})$. Let $H(\nu)$ denote a measure of information that satisfies null-uncertainty and concavity. Then the following two statements are equivalent*

1. *$d$ satisfies null-information, positivity, order-invariance, and is coupled with $H$.*

2. *$d$ is a Bregman divergence of $H$.*

We can further refine the results by focusing on distributions that can be expressed as some $g_q$ which is the convolution of the prior $q$ and DPP $g_\beta$. In other words, $q$ is some belief over $\beta$ and $g_\beta$ the model itself.

**Proposition E.5.** *For any $l$ and measure of information $d$ that satisfies null-information and order-invariance, let $H(g_q) = \int q(\beta)d(g_\beta, g_q)d\beta$. Then $H$ is coupled with $d$ and satisfies null-uncertainty.*

Before discussing the implications, we strengthen Proposition E.5 via the following corollary.

**Corollary E.6.** *For any $d$ that satisfies null-information, positivity, and order-invariance, there is a unique measure of uncertainty $H(g_q) = \int q(\beta)d(g_\beta, g_q)d\beta$ that satisfies null-uncertainty, concavity, and is coupled with d.*

These results show that once the measure of information $d$ is determined, not only can the parametric form of $H$ can be expressed explicitly, but the form itself is further unique, even in Bayesian regression settings where the response is continuous. With the properties on $d$ and $H$ laid out, we take a step back and demonstrate that the value of information `val` and the cost of uncertainty `cost` share similar properties, despite their original motivations as decision-making problems.

**Proposition E.7.** *Let `val` be the value of information and `cost` the cost of uncertainty of a contextual decision-making problem. We know (1) `val` satisfies null-information, positivity, and order-invariance, (2) `cost` satisfies null-uncertainty and concavity, and (3) `val` and `cost` are coupled.*

Proposition E.7 combined with Lemma E.4 show that the value of information is the Bregman divergence defined with respect to the cost of uncertainty in any contextual decision-making problem associated with $\mathcal{A}$, $u$, and $x$, showing that Bregman divergences are intrinsically linked to decision-making.

**Proposition E.8.** *For any model $g$ and measure of uncertainty $H$ that satisfies null-uncertainty, and concavity, letting $d$ be the Bregman divergence $H$ induces, there is some decision-making problem in the direction $x$ where the cost of uncertainty* $\mathtt{cost}(q, x) = H(g_q)$ *and the value of information* $\mathtt{val}(D; q, x) = d(g_p, g_q)$, *where $p$ is the posterior induced by $D$.*

Proposition E.8 stipulates that as long as the concave functional $H$ satisfies the conditions in Definition E.3, then the functional and its Bregman divergence correspond to the value of information and the cost of uncertainty in a decision-making problem.

We now discuss the proofs for Proposition 2.6, Proposition B.4, Theorem 2.7.

**Proof of Proposition 2.6.** The if direction, that is, any `val` must satisfy null-information, positivity, and order-invariance is proven by Proposition E.7. We then consider when `val` satisfies the properties. Since the definition does not limit the choice of possible data $D$, for any $p$ that can be induced by $q$, we can devise a corresponding dataset that induces $p$. Consequently, we can construct some measure of information $d$ via value of data `val`. By Corollary E.6, we can explicitly write out a corresponding $H$ coupled with $d$ and, by extension, a corresponding `cost` that is coupled with `val`. By Proposition E.8, `cost` and `val` are then the cost of information and value of uncertainty for some decision-making problems.

**Proof of Proposition B.4.** The if direction, that is, any `cost` must satisfy null-uncertainty and concavity is proven by Proposition E.7. We then consider when `cost` satisfies both properties. Note that `cost` can be written as some measure of uncertainty $H$ and, by Proposition E.8, there exists a corresponding $d$ and, consequently, a corresponding `val`.

**Proof of Theorem 2.7.** The only if direction, that is, `val` must be a Bregman divergence of $H$, is implied by Lemma E.4 and the fact that `val` and `cost` are coupled by Proposition E.7. The if direction is established by Corollary E.6, which shows the uniqueness of `cost` that couples with `val`, and Proposition E.8, which constructs a suitable decision-making problem for which `val` and `cost` are valid.

### E.1. Proof of Lemma E.4

We begin by showing the first statement implies the second.

Let $\Omega$ denote the space of probability distributions and $\tau \in \Delta(\Omega)$. Well-known result in Bayesian persuasion tells us that if $\mathbb{E}_{\mu \sim \tau}[\mu] = \nu$, then there must be a signaling scheme $\pi$ that induces the distribution $\tau$[9].

Since $\mathbb{E}_s[d(\nu(s), \nu)] = \mathbb{E}_s[H(\nu(s)) - H(\nu)]$, $\mathbb{E}_{\mu \sim \tau}[d(\mu, \nu) - H(\nu) + H(\mu)] = 0$ for all distributions $\tau$ such that $\mathbb{E}_{\mu \sim \tau}[\mu] = \nu$.

We now show $d(\mu, \nu) - H(\nu) + H(\mu)$ is linear in $\mu$ for any $\nu$. For convenience, we use the following shorthand notation

$$\gamma(\mu; \nu) = d(\mu, \nu) - H(\nu) + H(\mu).$$

Let $\mu$ be an arbitrary distribution that can be induced from $\nu$ by some signaling scheme and assume there exists some constant $C$ and measure $\Delta$ such that $\mu = \nu + C\Delta$, where $C$ is an arbitrary yet sufficiently large constant that ensures $\nu \pm \Delta$ are both valid distributions. Note that $C$ must exist for any $\Delta$ as $\mu$ can be written as the posterior of $\nu$ under some signaling scheme, implying that $\mu$ must be absolutely continuous with respect to $\nu$. Consider then two possible posteriors. Consider first $\tau_1$, which assigns probability mass $\frac{1}{2}$ to both $\nu + \Delta$ and $\nu - \Delta$. Since $\mathbb{E}_{\mu \sim \tau_1}[\mu] = \nu$, we have

$$\gamma(\nu + \Delta; \nu) = -\gamma(\nu - \Delta; \nu).$$

Then consider $\tau_2$, which assigns probability $\frac{1}{1+C}$ and $\frac{C}{1+C}$ to $\nu + C\Delta$ and $\nu - \Delta$, respectively. Since the expectation of the distribution is again $\nu$, We have

$$C\gamma(\nu - \Delta; \nu) + \gamma(\nu + C\Delta; \nu) = 0 \implies \gamma(\nu + C\Delta; \nu) = -C\gamma(\nu - \Delta; \nu) = C\gamma(\nu + \Delta; \nu).$$

Take $\|\Delta\| \to 0$ and we know $\gamma(\nu + C\Delta; \nu) = C\gamma(\nu + \Delta; \nu)$ for all $C > 0$ and all directions $\Delta$ may take. Therefore, $\gamma(\mu; \nu)$ must be linear in $\mu$ and thus can be written as an affine function. Moreover, note that $\gamma(\nu; \nu) = 0$ by direct calculation.

---

[9]Since $\mathbb{E}_{\mu \sim \tau}[\mu] = \nu$ and probability measures are non-negative, $\mu$ must be absolutely continuous w.r.t. $\nu$. We then take $\pi = \frac{\mu}{\nu}$, which is well defined by absolute continuity and integrates to 1 by the fact that $\mathbb{E}_{\mu \sim \tau}[\mu] = \nu$.

There then must exist a functional $f(\nu)$ such that

$$d(\mu, \nu) - H(\nu) + H(\mu) = \gamma(\mu; \nu) = \langle f(\nu), \mu \rangle \implies d(\mu, \nu) = H(\nu) - H(\mu) + \langle f(\nu), \mu - \nu \rangle.$$

Moreover, by positivity, we know that

$$d(\mu, \nu) = H(\nu) - H(\mu) + \langle f(\nu), \mu - \nu \rangle \geq 0$$

for all pairs of distributions. By definition of superdifferentials, the bound necessarily implies $f(\nu)$ must be a superdifferential of $H$, and hence $d(\mu, \nu)$ is the Bregman divergence of $H$. In other words, the first statement implies the second.

We now show the second statement implies the first. Let $\nabla H$ denote an arbitrary and fixed superdifferential of $H$ and $d(\mu, \nu) = H(\nu) - H(\mu) + \langle \nabla H(\nu), \mu - \nu \rangle$. By the property of superdifferential and the fact that $H$ is concave, $d$ satisfies positivity. Note that for any signaling scheme $\pi_s : \mathcal{Y} \to \mathcal{S}$, we have $\mathbb{E}_s[\nu(s)] = \nu$, and thus

$$\mathbb{E}_s[d(\nu(s), \nu)] = \mathbb{E}_s[H(\nu) - H(\nu(s))] + \langle \nabla H(\nu), \mathbb{E}_s[\nu(s) - \nu] \rangle = \mathbb{E}_s[H(\nu) - H(\nu(s))],$$

showing $d$ and $H$ are coupled. Additionally, $d$ easily satisfies null-information, as $d(\nu, \nu) = H(\nu) - H(\nu) + \langle \nabla H(\nu), \nu - \nu \rangle = 0$. Thus, all that remains is to show $d$ satisfies order invariance. Recalling we have already shown $d$ and $H$ are coupled, for any signals $s$ and $s'$, we have

$$
\begin{aligned}
\mathbb{E}_{s,s'}&[d(\nu(s), \nu) + d(\nu(s \cup s'), \nu(s))] \\
&= \mathbb{E}_{s,s'}[H(\nu) - H(\nu(s)) + H(\nu(s)) - H(\nu(s \cup s'))] \\
&= \mathbb{E}_{s,s'}[d(\nu(s'), \nu) + d(\nu(s \cup s'), \nu(s'))],
\end{aligned}
\tag{5}
$$

thereby completing the proof. $\qquad \square$

## E.2. Proof of Proposition E.5

Consider two direct signals $\pi_{s_1}$ and $\pi_{s_2}$ where with a slight abuse of notation we let $\pi_{s_2}$ denote the fully informative signaling scheme, that is,

$$\pi_{s_2}(\beta') = \mathbb{1}\{\beta = \beta'\},$$

which directly tells the buyer the underlying parameter $\beta$. We thus have

$$\mathbb{E}_{s_2}[d(g_{q(s_2)}, g_q)] = \mathbb{E}[d(g_\beta, g_q)] = \int q(\beta) d(g_\beta, g_q) d\beta = H(g_q).$$

Consider first the case where $s_1$ is observed *after* $s_2$. As $\pi_{s_1}$ cannot further alter the probability distribution, due to $\pi_{s_2}$ being fully informative, we know $q(s_1 \cap s_2) = q(s_2)$. Using the fact that $d$ satisfies null-information, we have $\mathbb{E}_q[d(g_{q(s_1 \cap s_2)}, g_{q(s_2)})] = 0$ for all possible realizations of the signals $s_1, s_2$, which in turn implies

$$\mathbb{E}_q[d(g_{q(s_2)}, g_q) + d(g_{q(s_1 \cap s_2)}, g_{q(s_2)})] = H(g_q).$$

We now consider the case where the fully informative $s_2$ is observed *after* $s_1$. Again exploiting the fact that $q(s_1 \cap s_2) = q(s_2)$, for any realized signal $s_1$

$$\mathbb{E}_q[d(g_{q(s_1 \cap s_2)}, g_{q(s_1)})] = \mathbb{E}_q[d(g_\beta, g_{q(s_1)})] = H(g_{q(s_1)}).$$

As $d$ satisfies order-invariance, we have

$$
\begin{aligned}
\mathbb{E}_{s_1, s_2, \beta}&[d(g_{q(s_1)}, g_q) + d(g_{q(s_1 \cap s_2)}, g_{q(s_1)})] = \mathbb{E}_{s_1, s_2}[\mathbb{E}_q[d(g_{q(s_2)}, g_q) + d(g_{q(s_1 \cap s_2)}, g_{q(s_2)})]] \\
&= \mathbb{E}_{s_1, s_2}[H(g_q)] = H(g_q).
\end{aligned}
$$

By direct expansion and linearity of expectation, we also know that

$$\mathbb{E}_{s_1, s_2, \beta}[d(g_{q(s_1)}, g_q) + d(g_{q(s_1 \cap s_2)}, g_{q(s_1)})] = \mathbb{E}_{s_1}[d(g_{q(s_1)}, g_q)] + \mathbb{E}_{s_1}[H(g_{q(s_1)})].$$

In other words, for any signaling scheme $\pi_{s_1}$ and signal $s_1$, we have

$$\mathbb{E}_{s_1}[d(g_{q(s_1)}, g_q)] = \mathbb{E}_{s_1}[H(g_q) - H(g_{q(s_1)})],$$

which shows that $d$ and $H$ are coupled. $\qquad \square$

### E.3. Proof of Corollary E.6

We first show that the proposed $H$ satisfies concavity. Note that for any parameter $\beta$ we have

$$H(g_\beta) = d(g_\beta, g_\beta) = 0,$$

since $d$ satisfies null-information. For concavity, let $q_1$ and $q_2$ be two arbitrary and fixed measures over the distribution of $\beta$ and let $c \in (0, 1)$ be an arbitrary scalar. Let $q = cq_1 + (1 - c)q_2$ be the convex combination of $q_1$ and $q_2$. It is easy to see that there exists a signaling scheme $\pi_{s_1}$ that induces $q_1$ with probability $c$ and $q_2$ with probability $1 - c$. Applying Proposition E.5, we know $H$ is coupled with $d$, and thus

$$\mathbb{E}_{s_1}[H(g_q) - H(g_{q(s_1)})] = \mathbb{E}_{s_1}[d(g_{q(s_1)}, g_q)] \geq 0$$

by positivity of $d$. We then know $H$ is concave by noting $\mathbb{E}_{s_1}[H(g_{q(s_1)})] = cH(g_{q_1}) + (1 - c)H(g_{q_2})$. Applying Proposition E.5 again, we know that $H$ further satisfies null-uncertainty and is coupled with $d$.

All that remains is to show that $H$ is unique. Again consider some fully informative signal $\pi_{s_2}$. Let $\widehat{H}$ be an arbitrary measure of information that satisfies null-uncertainty, concavity, and is coupled with $d$. By null-uncertainty of $\widehat{H}$, for any distribution $q$

$$\widehat{H}(g_q) - \mathbb{E}_{s_2}[\widehat{H}(g_{q(s_2)})] = \widehat{H}(g_q).$$

Moreover, as $\widehat{H}$ and $d$ are coupled

$$\widehat{H}(g_q) - \mathbb{E}_{s_2}[\widehat{H}(g_{q(s_2)})] = \mathbb{E}_{s_2}[d(g_{q(s_2)}, g_q)] = \int q(\beta)d(g_\beta, g_q)d\beta = H(g_q).$$

In other words, the proposed $H$ is unique as $\widehat{H}(g_q) = H(g_q)$ for arbitrary $q$, completing the proof. $\qquad\square$

### E.4. Proof of Proposition E.7

We begin by proving the second statement. Showing `cost` satisfies null-uncertainty is straightforward as we directly plug in the definition. Let $q_1$ and $q_2$ denote two arbitrary and fixed distributions over $\beta$ and let $q = cq_1 + (1 - c)q_2$ be an arbitrary convex combination of the two. Directly writing out the expectations, we know $\mathbb{E}_{g_q}[u(a; y)] = c\mathbb{E}_{g_{q_1}}[u(a; y)] + (1 - c)\mathbb{E}_{g_{q_2}}[u(a; y)]$ for any action $a$ and thus

$$c\mathbb{E}_{g_{q_1}}[u(a^*(q_1); y)] + (1 - c)\mathbb{E}_{g_{q_2}}[u(a^*(q_2); y)]$$
$$\geq c\mathbb{E}_{g_{q_1}}[u(a^*(q); y)] + (1 - c)\mathbb{E}_{g_{q_2}}[u(a^*(q); y)]$$
$$= \mathbb{E}_{g_q}[u(a^*(q); y)].$$

Moreover, also by directly writing out the expectations,

$$c\mathbb{E}_{g_{q_1}}[u(a^*(\delta_\beta); y)] + (1 - c)\mathbb{E}_{g_{q_1}}[u(a^*(\delta_\beta); y)] = \mathbb{E}_{g_q}[u(a^*(\delta_\beta); y)].$$

Combining the two shows that $\text{cost}(q)$ is concave in $q$, completing the proof for the second property.

We then show the third property holds. Let $q$ be some arbitrary belief over $\beta$ and, for convenience, $D$ an arbitrary dataset given. Let $p$ be the posterior induced by the dataset. We have

$$\mathbb{E}_D[\text{cost}(q, x) - \text{cost}(p, x)]$$
$$= \mathbb{E}_D\Big[\mathbb{E}_{g_q}[u(a^*(\delta_\beta); y) - u(a^*(q); y)] - \mathbb{E}_{g_p}[u(a^*(\delta_\beta); y) - u(a^*(p); y)]\Big].$$

Notice that $\mathbb{E}_D[p] = q$ by Bayes' theorem. Therefore

$$\mathbb{E}_D\left[\mathbb{E}_{g_q}[u(a^*(\delta_\beta); y)] - \mathbb{E}_{g_p}[u(a^*(\delta_\beta); y)]\right] = 0,$$

and thus

$$\mathbb{E}_D[\text{cost}(q, x) - \text{cost}(p, x)] = \mathbb{E}_D\left[\mathbb{E}_{g_p}[u(a^*(p); y)] - \mathbb{E}_{g_q}[u(a^*(q); y)]\right].$$

We now turn our attention back to val. Notice that

$$\mathbb{E}_D[\mathtt{val}(D; q, x)] = \mathbb{E}_D[\mathbb{E}_{g_p}[u(a^*(p); y) - u(a^*(q); y)]].$$

Focus on the term $u(a^*(q); y)$. As the optimal action is taken over the distribution $q$, its value for any specific $y$ is independent of the realized value of $D$. Therefore, again using Bayes' rule to derive, we have

$$\mathbb{E}_D[\mathbb{E}_{g_p}[u(a^*(q); y)]] = \mathbb{E}_q[u(a^*(q); y)],$$

recalling that $q$ is the prior distribution. Noting that taking the expectation of the term above over $D$ does not affect its value. Therefore

$$\begin{aligned}
\mathbb{E}_D[\mathtt{val}(D; q, x)] &= \mathbb{E}_D[\mathbb{E}_{g_p}[u(a^*(p); y) - u(a^*(q); y)]] \\
&= \mathbb{E}_D\left[\mathbb{E}_{g_p}[u(a^*(p); y)] - \mathbb{E}_{g_q}[u(a^*(q); y)]\right] \\
&= \mathbb{E}_D[\mathtt{cost}(q, x) - \mathtt{cost}(p, x)]
\end{aligned}$$

completing our proof of the third property, as we have now shown val and cost are coupled.

Finally we focus on the first property. Null-information and positivity are straightforward by definition of val. Furthermore, as val and cost are coupled, similar to Equation (5), we know that val also satisfies order invariance, completing the proof of the first property. □

## E.5. Proof of Proposition E.8

Let the action set $\mathcal{A}$ be a subset of all possible distributions over $\mathcal{Y}$ and consider the following utility function

$$u(a; y) = \begin{cases} H(a) - \langle \nabla H(a), a \rangle + \langle \nabla H(a), \delta_y \rangle & \text{if } a \text{ is strictly positive at } y \\ -\infty & \text{otherwise,} \end{cases}$$

where $\delta_y$ is the Dirac measure defined at $y$ and $\nabla H$ follows its definition in Definition 2.2. An immediate consequence of the construction is that for any model $l$ and parameter $\beta$

$$\begin{aligned}
\mathbb{E}_{y \sim g_\beta}[u(a; y)] &= \int u(a; y) g_\beta(y) dy \\
&= (H(a) - \langle \nabla H(a), a \rangle) \int g_\beta(y) dy + \int \langle \nabla H(a), \delta_y \rangle \cdot g_\beta(y) dy \\
&= H(a) - \langle \nabla H(a), a \rangle + \langle \nabla H(a), g_\beta(y) \rangle \\
&= H(a) + \langle \nabla H(a), g_\beta - a \rangle + H(g_\beta) \\
&= d(g_\beta, a),
\end{aligned}$$

where for the second to last equality we use the fact that $H$ satisfies null-uncertainty and the last equality leverages the fact that $d$ is the Bregman divergence with respect to $H$. We then know that for any $q$

$$\begin{aligned}
\mathbb{E}_{y \sim g_q}[u(a; y)] &= \int u(a; y) g_q(y) dy \\
&= \int \left( \int q(\beta) g_\beta(y) d\beta \right) u(a; y) dy \\
&= \int q(\beta) \left( \int g_\beta(y) u(a; y) dy \right) d\beta \\
&= \int q(\beta) d(g_\beta, a) d\beta \\
&= \mathbb{E}_q[d(g_\beta, a)].
\end{aligned}$$

Consider then two beliefs over $\mathcal{Y}$, induced by beliefs over $\beta$, denoted $p$ and $q$, such that $q$ is absolutely continuous with respect to $p$. The actions $p$ and $q$ then satisfy

$$
\begin{aligned}
\mathbb{E}_{g_p}[u(p; y) - u(q; y)] &= \mathbb{E}_p[d(g_\beta, p) - d(g_\beta, q)] \\
&= \mathbb{E}_p[H(g_q) - H(g_\beta) + \langle \nabla H(g_q), g_\beta - q \rangle \\
&\quad - H(g_p) + H(g_\beta) - \langle \nabla H(g_p), g_\beta - p \rangle] \\
&= \mathbb{E}_p[H(g_q) - H(g_p) + \langle \nabla H(g_q), p - q \rangle] \\
&= H(g_q) - H(g_p) + \langle \nabla H(g_q), p - q \rangle = d(p, q) \geq 0,
\end{aligned}
$$

where for the third equality we use the linearity of $\langle \nabla H(g_p), g_\beta - p \rangle$ in $g_\beta$ and recall that $p = \mathbb{E}_p[g_\beta]$ by definition. In other words, for any belief $p$, the buyer's utility can only be harmed by moving from $p$ to some belief $q$ that is absolutely continuous with respect to $p$. Moreover, by the construction of $u(a; y)$, we know that it is impossible for any $q$ that is not absolutely continuous with respect to $p$ to be near optimal, as otherwise there exists some $y$ for which $u(q; y) = -\infty$.

Consequently, for any belief $p$, the action $p$ maximizes the expected utility $\mathbb{E}_{g_p}[u(a; y)]$. Recalling the definition of the value of information, we thus have

$$
\texttt{val}(D; q, x) = \mathbb{E}_{g_p}[u(a^*(p); y) - u(a^*(q); y)] = d(p, q),
$$

where $p$ is the posterior induced by $D$. In other words, $d(p, q)$ is a value of information for the contextual decision-making problem associated with $\mathcal{A}$, $x$, and $u$. By Proposition E.7, it is then coupled with the cost of uncertainty of the same decision-making problem. Since $H$ is coupled with $d$ and satisfies null-uncertainty and concavity, by Corollary E.6 it is unique, and thus $H$ is the cost of uncertainty for the problem with $\mathcal{A}$, $x$, and $u$. □

# F. Omitted Proof in Section 3

### F.1. Proof of Theorem 3.1

We first state that entropy leads to a qualified candidate of the value function, and then we derive its closed form.

We focus on $\mathbb{E}[y[x]] = \langle x, \beta \rangle$. There are two reasons we use it rather than $y$. First, the composition of $u(\cdot; \cdot)$ and the expectation operator won't influence the existence of $u$ and then the existence of $V(\cdot; \cdot, \cdot)$ as $\epsilon$ is white noise. Second, in reality, people always focus on $f_\beta(x)$ rather than the one with uninformative noise $\epsilon$. So, the entropy of predicting $f_\beta(x)$, namely, $\langle x, \beta \rangle$ makes more sense in practice.

We are now going to show that entropy can generate a valid value function. Note that we can view $g$ as a signal transforming our belief over $\beta$. The model $l$ for $\mathbb{E}[y[x]]$ is then $\mathbb{E}[y[x]] = \langle \beta, x \rangle$. Consider the measure of information $d(p, q) = D_{\mathrm{KL}}(p\|q)$.

By Corollary E.6, the unique measure of uncertainty corresponding to $d$ is

$$
H(g_q) = \int q(\beta) D_{\mathrm{KL}}(g_\beta \| g_q) d\beta.
$$

In this case, letting $g_q$ denote the prior belief over $\mathbb{E}[y[x]] = \langle x, \beta \rangle$ and associating with the posterior $p = p(\cdot \mid q, D)$, by definition of coupling between $H$ and $d$, we have

$$
\begin{aligned}
\mathbb{E}_{D \sim g_q}[d(g_p, g_q)] &= \mathbb{E}_{D \sim g_q}[H(g_q) - H(g_p)] \\
&= \mathbb{E}_{D \sim g_q}\left[\int q(\beta) D_{\mathrm{KL}}(g_\beta, g_q) d\beta\right] - \mathbb{E}_{D \sim g_q}\left[\int p(\beta) D_{\mathrm{KL}}(g_\beta, g_p) d\beta\right].
\end{aligned}
$$

We focus on the terms involving $q$ and $g_q$, as the case for $p$ and $g_p$ is similar. Notice that

$$\int q(\beta) D_{\mathrm{KL}}(g_\beta, g_q) d\beta$$

$$= -\int\int q(\beta) l(\mathbb{E}[y[x]]; x, \beta) \log(g_q(\mathbb{E}[y[x]])) d(\mathbb{E}[y[x]]) d\beta$$

$$+ \int\int q(\beta) l(\mathbb{E}[y[x]]; x, \beta) \log(l(\mathbb{E}[y[x]]; x, \beta)) d(\mathbb{E}[y[x]]) d\beta$$

$$= h(f_q(x)) + \mathbb{E}_{\beta \sim q}\left[\int l(\mathbb{E}[y[x]]; x, \beta) \log(l(\mathbb{E}[y[x]]; x, \beta)) d(\mathbb{E}[y[x]]) d\beta\right],$$

where the last line comes from the direct observation that

$$h(f_q(x)) = -\int\int q(\beta) l(\mathbb{E}[y[x]]; x, \beta) \log(g_q(\mathbb{E}[y[x]])) d(\mathbb{E}[y[x]]) d\beta.$$

Similarly, recalling the definition of $h(f_p(x))$, we know that

$$\mathbb{E}_{D \sim g_q}[d(g_p, g_q)] = \mathbb{E}_{D \sim g_q}[h(f_q(x)) - h(f_p(x))]$$

$$+ \mathbb{E}_{\beta \sim q}\left[\int l(\mathbb{E}[y[x]]; x, \beta) \log(l(\mathbb{E}[y[x]]; x, \beta)) d(\mathbb{E}[y[x]]) d\beta\right]$$

$$- \mathbb{E}_{\beta \sim p}\left[\int l(\mathbb{E}[y[x]]; x, \beta) \log(l(\mathbb{E}[y[x]]; x, \beta)) d(\mathbb{E}[y[x]]) d\beta\right].$$

Observe that in the model we defined, $\int l(\mathbb{E}[y[x]]; x, \beta) \log(l(\mathbb{E}[y[x]]; x, \beta)) d(\mathbb{E}[y[x]])$ is independent of the value of $\beta$, and thus the latter two terms cancel out. Moreover, as we will soon discuss in the proof of Theorem 3.1, in the special case of Gaussian distribution, $h(f_p(x))$ is affected by only the feature matrix, but not necessarily the realized labels. In other words, when $D = \{(x_i, y_i)\}_{i=1}^n$, $\mathbb{E}_{D \sim g_q} h(f_p(x))$ only depends on $\{x_1, ..., x_n\}$ and we have that

$$h(f_q(x)) - h(f_p(x)) = d(g_p, g_q),$$

where $d(g_p, g_q)$ is the measure of information defined in Definition E.1 that serves as the value of information of some decision-making problem. Therefore, we can have the following formal definition which coincides with information gain in Bayesian regression.

**Definition F.1** (Information Gain). For a buyer type $x$, a valid value, i.e., the information gain from $g$ is

$$\mathtt{V}(g; q, x) := \mathbb{E}_D[h(f_q(x)) - h(f_p(x))],$$

where $h(f_q(x))$ and $h(f_p(x))$ are the entropy of his prior and posterior distributions over $\mathbb{E}[y[x]]$ and $p$ depends on both $q$ and $D$.

Note that $\mathtt{V}(\cdot; \cdot, \cdot)$ is exactly the expected value of $D$ to the buyer when he makes the purchase. The fact that $\mathtt{V}(\cdot; \cdot, \cdot)$ is the buyer's ex-ante value of data is crucial, as it is impossible for the buyer to ascertain exactly the data's value to him prior to observing $D$ and updating his belief. It justifies the choice of information gain and we list its detailed concept and closed form under Bayesian linear regression which provides some convenience for optimal mechanism design in Sections 4.1 and 4.2.

Information gain is a concept commonly used in statistical learning literature, whose applications range from experiment design (Beck et al., 2018), to bandits (Vakili et al., 2021), and to feature selection (Azhagusundari et al., 2013). As we show in the sequel, it also has the desirable property that it is invariant to scaling of the feature vector $x$, ensuring that the value of data depends on $x$ only through its direction.

As we have shown differential entropy yields a valid value function, it's time to derive the concrete form in Theorem 3.1. We first state the following lemma.

**Lemma F.2** (Label's Prior Entropy). *The entropy of the buyer's prior distribution over $\mathbb{E}[y[x]]$ is*

$$h(f_q(x)) = \frac{1}{2}\log(|x\Sigma_q x^T|) + \frac{d}{2}(1 + \log(2\pi)),$$

*where we use $\Sigma_q$ to denote the covariance of the prior distribution over $\beta$.*

*Proof.* We begin by stating the following basic facts on the entropy of Gaussian random vectors.

*Fact* F.3 (Entropy of Gaussian Random Vectors (Example 8.1.2 Thomas & Joy (2006))). For any Gaussian random vector $Z \in \mathbb{R}^d$ where $Z \sim \mathcal{N}(m, \Sigma)$, it holds that

$$h(Z) = \frac{1}{2}\log(|\Sigma|) + \frac{d}{2}(1 + \log(2\pi)).$$

We know that $\mathbb{E}[y[x]] = \langle x, \beta \rangle$ where $\beta$ enjoys prior variance $\Sigma_q$. As we recall that $\mathbb{E}[y[x]]$ integrates over a zero-mean random noise $\mathcal{N}(0, \sigma^2)$, the prior belief that the buyer holds over $\mathbb{E}[y[x]]$ is that it has variance $x\Sigma_q x^T$. Applying Fact F.3 completes the proof. $\square$

With Lemma F.2 and Fact F.3 in hand, we can similarly derive $h(f_p(x))$ by substituting the prior variance with posterior variance. Then Definitions C.1 and F.1 lead to Theorem 3.1 directly. $\square$

### F.2. Proof of Corollary 3.2

Since the prior is $\mathcal{N}(\mu_q, \Sigma_q)$, after observing $\{(x_i, y_i)\}_{i=1}^n$, with calculations using Equation (2), we know the variance matrix of the posterior of $\beta$ is $(\Sigma_q^{-1} + X^T\Sigma^{-1}X)^{-1}$. To be specific,

$$p(\beta \mid q, D) \propto \exp(-\frac{1}{2}\beta^T\Sigma_q^{-1}\beta) * \exp(-\frac{1}{2}(Y - X\beta)^T\Sigma^{-1}(Y - X\beta)).$$

Note that it's independent of $\{y_1, ..., y_n\}$ so we can get rid of the expectation over them. Then, Equation (4) yields Corollary 3.2 immediately. $\square$

Note that $\text{val}(\cdot; \cdot, \cdot)$ here only depends on the design matrix $X$ but not the realized $y$. Hence, we can get rid of $\mathbb{E}_{D \sim g \circ q}$ in such a case.

# G. Omitted Proof in Section 4

## G.1. Omitted Proof in Section 4.1

Readers may wonder why we use $V(g^n[x]; q, x)$ as the benchmark revenue. We assume without loss of generality, the buyer with type $x$ prefers the associated data-generating process, ceteris paribus. It equals to $\frac{n + \sigma(\widehat{x})^2}{n + \sigma(x)^2} \geq \langle x, \widehat{x} \rangle^2$. Taking union bound over $n$, a sufficient condition is $\frac{\sigma(\widehat{x})}{\sigma(x)} \geq |\langle x, \widehat{x} \rangle|$. Besides, in practice, buyers may lack information on $\sigma$ and it's only revealed after data production processes. Therefore, there is no motivation for buyers to misreport based on $\sigma$. Note that this assumption is not necessary but only for a concise presentation. Otherwise, there exists a mapping $\varphi : \mathbb{R}^d \to \mathbb{R}^d$ such that $\varphi(x)$ is type-$x$ buyer's favorite direction. Composing $\varphi(\cdot)$ with the instrumental value function $V(\cdot, \cdot)$ yields an equivalent mechanism design problem.

Additionally, from Corollary 3.2, we know that only the design matrix will influence $V$ rather than response variable values.

### G.1.1. PROOF OF LEMMA D.1

Since we have Theorem 3.1, we only need to calculate the posterior distribution of $\beta$. For data $g[\widehat{x}] = (\widehat{x}, \{\widehat{y}_i\}_{i=1}^n)$, using Bayesian Formula, it holds that

$$\mathbb{P}(\beta_p \mid \mathcal{N}(0, I_d), (\widehat{x}, \{\widehat{y}_i\}_{i=1}^n)) \propto \mathcal{N}(0, I_d) * \mathbb{P}(\widehat{x}, \{\widehat{y}_i\}_{i=1}^n \mid \beta)$$
$$\propto \exp(-\frac{1}{2}\|\beta\|^2) * \exp(-\sum_{i=1}^n \frac{1}{2}\frac{(\widehat{y}_i - \widehat{x}^T\beta)^2}{\sigma(\widehat{x})^2 + \delta(\widehat{x})^2})$$
$$\propto \exp(-\frac{1}{2}(\beta^T(I_d + \frac{n\widehat{x}\widehat{x}^T}{\sigma(\widehat{x})^2 + \delta(\widehat{x})^2})\beta)),$$

considering only the quadratic terms of $\beta$. It is because that $\epsilon$ and $\epsilon'$ are independent and $\text{Var}(\epsilon + \epsilon') = \sigma^2 + \delta^2$. From now on, we omit the prior $q = \mathcal{N}(0, I_d)$ when easy to infer from the context for simplicity.

Therefore, we have that the variance of the posterior distribution of $\beta$ is

$$\text{Var}(\beta_p \mid g[\widehat{x}]) = \Sigma_p = (I_d + \frac{n\widehat{x}\widehat{x}^T}{\sigma(\widehat{x})^2 + \delta(\widehat{x})^2})^{-1},$$

and Theorem 3.1 leads to what we need. $\qquad\square$

### G.1.2. PROOF OF LEMMA D.2

Now, we are going to prove Lemma D.2. With IC constraint, it holds that

$$\text{V}(x, x) - t(x) \geq \text{V}(x, \widehat{x}) - t(\widehat{x}) \text{ for all } x, \widehat{x}.$$

Rearranging the algebra items leads to $t(\widehat{x}) - t(x) \geq \text{V}(x, \widehat{x}) - \text{V}(x, x)$. Now, we change the placement of $x$ and $\widehat{x}$, it holds that $t(\widehat{x}) - t(x) \leq \text{V}(\widehat{x}, \widehat{x}) - \text{V}(\widehat{x}, x)$. Combining the two parts, we have

$$\text{V}(x, \widehat{x}) - \text{V}(x, x) \leq t(\widehat{x}) - t(x) \leq \text{V}(\widehat{x}, \widehat{x}) - \text{V}(\widehat{x}, x).$$

Then, dividing by $\widehat{x} - x$ and letting $\widehat{x}$ converge to $x$ results in $\nabla t(x) = \nabla_y \text{V}(x, y) \mid_{y=x}$.

Now, we calculate the gradient of $\text{V}(\cdot, \cdot)$. We have the following equations.

$$
\begin{aligned}
\nabla_y \text{V}(x, y) \mid_{y=x} &= -\frac{1}{2} \nabla_y \log(x^T (I + \frac{nyy^T}{\sigma(y)^2 + \delta(y)^2})^{-1} x) \\
&= -\frac{1}{2} \nabla_y \log(1 - \frac{x^T yy^T x}{(\sigma(y)^2 + \delta(y)^2)/n + 1}) \mid_{y=x} \\
&= -\frac{1}{2}(1 - \frac{x^T yy^T x}{\frac{\sigma(y)^2 + \delta(y)^2}{n} + 1})^{-1} [\frac{2x^T yx}{\frac{\sigma(y)^2 + \delta(y)^2}{n} + 1} - \frac{2\frac{\sigma(y)}{n}\nabla\sigma(y) + 2\frac{\delta(y)}{n}\nabla\delta(y)}{(\frac{\sigma(y)^2 + \delta(y)^2}{n} + 1)^2}] \mid_{y=x} \\
&= -\frac{1}{2} \frac{\frac{\sigma(x)^2 + \delta(x)^2}{n} + 1}{\frac{\sigma(x)^2 + \delta(x)^2}{n}} (\frac{2x}{\frac{\sigma(x)^2 + \delta(x)^2}{n} + 1} + \frac{2\sigma(x)/n\nabla\sigma(x) + 2\delta(x)/n\nabla\delta(x)}{((\sigma(x)^2 + \delta(x)^2)/n + 1)^2}).
\end{aligned}
$$

The first equation comes from the definition $\text{V}(\cdot, \cdot)$ while the third equation comes from the derivation of multi-variant functions. When we replace $y$ with $x$, we have the last equation since $\|x\|^2 = x^T x = 1$. As for the second equation, we need the following lemma.

**Lemma G.1** (Sherman-Morrison-Woodbury Formula (Sherman & Morrison, 1950)). *Let $A$ and $B$ be nonsingular $m \times m$ and $n \times n$ matrices, respectively, and let $U$ be $m \times n$ and $V$ be $n \times m$. Then*

$$(A + UBV)^{-1} = A^{-1} - A^{-1}UB(B + BVA^{-1}UB)^{-1}BVA^{-1}.$$

Taking $A = I, U = y, V = y^T$ and $B = \frac{n}{\sigma(y)^2 + \delta(y)^2}$ in Lemma G.1 leads to the second equation.

Since $x \in \{x : \|x\| = 1\}$, it holds that $x \cdot dx = 0$. Then, it holds that

$$
\begin{aligned}
t(x) &= C + \int_{x_0}^x \nabla_y \text{V}(s, y) \mid_{y=s} \cdot ds \\
&= C + \int_{x_0}^x -\frac{1}{2} \frac{(\sigma(s)^2 + \delta(s)^2)/n + 1}{(\sigma(s)^2 + \delta(s)^2)/n} \frac{2\sigma(s)/n\nabla\sigma(s) + 2\delta(s)/n\nabla\delta(s)}{((\sigma(s)^2 + \delta(s)^2)/n + 1)^2} \cdot ds \\
&= C + \frac{1}{2} \log(\frac{(\sigma(x)^2 + \delta(x)^2)/n + 1}{(\sigma(x)^2 + \delta(x)^2)/n}),
\end{aligned}
$$

with a little abuse of the constant $C$.

The first equation is straightforward however we need to check if it's independent of the path of integration. The second and third equations are some calculations. It's easy to see that the construction of $t(x)$ satisfies the condition in (Jehiel et al., 1999) that the field is conservative. Therefore, the value is irrelevant to the path of integration and the payment rule is well-defined. The proof suggests that our method has greater verifiability compared to other methods, such as cycle monotonicity (Lavi & Swamy, 2007), as well. $\qquad\square$

G.1.3. PROOF OF THEOREM 4.1

From Lemma D.2, we know that the form of optimal payment rule is $t(x) = C + \frac{1}{2} \log(\frac{\sigma(x)^2 + \delta(x)^2 + n}{\sigma(x)^2 + \delta(x)^2})$.

Now, we need to calculate the indicator function of the IR constraint that

$$\mathbb{1}(C + \frac{1}{2} \log(\frac{\sigma(x)^2 + \delta(x)^2 + n}{\sigma(x)^2 + \delta(x)^2}) + \frac{1}{2} \log(x^T(I + \frac{nxx^T}{\sigma(x)^2 + \delta(x)^2})^{-1}x) \leq 0) = \mathbb{1}(C \leq 0).$$

It is because that

$$x^T(I + \frac{nxx^T}{\sigma(x)^2 + \delta(x)^2})^{-1}x = x^T(I - \frac{n}{\sigma(x)^2 + \delta(x)^2 + n}xx^T)x$$

$$= 1 - \frac{n}{\sigma(x)^2 + \delta(x)^2 + n}$$

$$= \frac{\sigma(x)^2 + \delta(x)^2}{\sigma(x)^2 + \delta(x)^2 + n}.$$

The first equation comes from Lemma G.1 with $A = I, U = x, V = x^T$ and $B = \frac{n}{\sigma(x)^2 + \delta(x)^2}$. The second equation comes from $\|x\| = 1$ while the last equation is from simple algebra calculation.

Therefore, the optimal $C$ is zero for any $x$ and the payment rule is exactly

$$t(x) = \frac{1}{2} \log(\frac{\sigma(x)^2 + \delta(x)^2 + n}{\sigma(x)^2 + \delta(x)^2}).$$

As for $\delta(\cdot)$, we will show that the optimal solution is 0 too. Since $\log(\frac{x+n}{x})$ is a decreasing function with respect to $x$ for any fixed $n$, taking $\delta(\cdot) = 0$ leads to the optimality.

Then, for any type $x$, adding no noise and charging $\frac{1}{2} \log(\frac{n+\sigma(x)^2}{\sigma(x)^2})$ are the optimal solution to the optimization problem D.1 if the distribution of type is $\delta_x$ where $\delta_{(\cdot)}$ is the Dirac delta function. However, since the solution is optimal to any type $x$, it's also the optimal solution to the general optimization problem D.1 as long as satisfying the IC constraint which is guaranteed by Lemma D.2.

Combining the above parts leads to our theorem and finishes our proof. $\qquad\square$

## G.2. Omitted Proof in Section 4.2

G.2.1. PROOF OF THEOREM 4.2

Recall that the value of the data $g[\hat{x}]$ is

$$\mathtt{V}(x, \hat{x}) = \frac{1}{2} \log(|x \operatorname{Var}(\beta_q)x^T|) - \frac{1}{2} \log\left(|x \operatorname{Var}(\beta_p \,|\, g[\hat{x}])x^T|\right).$$

Similar to the argument presented in Section 4.1, we make the assumption that the variance of the prior belief regarding $\beta$ is $I_d$, without loss of generality. We also assume that the seller can attain the first-best revenue and we will demonstrate a contradiction in due course. To achieve the first-best revenue, since the buyer reveals their report after the seller produces the data, the seller must disclose all data in some condition for any report $\hat{x}$ in order to prevent loss of information and charge the corresponding information gain. For instance, let us assume that the type is represented by a two-dimensional vector denoted by $x = (a, b)^T$, where $a, b \in [1, 2]$. Hence, each data point assists in predicting each component of $\beta$. The first-best revenue implies that the payment rule is precisely $t(\hat{x}) = \mathtt{V}(\hat{x}, \hat{x})$ and the buyer will derive no consumer surplus.

Now, we are ready to calculate the exact form of $t(\cdot)$. We first calculate the posterior distribution of $\beta$. It holds that

$$\mathbb{P}(\beta \mid \mathcal{N}(0, I_d), g^X) \propto \mathcal{N}(0, I_d) * \mathbb{P}(X, Y \mid \beta)$$
$$\propto \exp(-\frac{1}{2}\|\beta\|^2) * \exp(-\frac{1}{2}(Y - X\beta)^T\Sigma^{-1}(Y - X\beta))$$
$$\propto \exp(-\frac{1}{2}\|\beta\|^2) * \exp(-\frac{1}{2}\beta^T X^T \Sigma^{-1} X\beta)$$
$$\propto \exp(-\frac{1}{2}\beta^T(I + X^T\Sigma^{-1}X)\beta)$$

The first line comes from the Bayesian formula and the following lines are simple algebra calculations. Recall that $\Sigma$ is a mapping from $X$. Therefore, it holds that $\mathrm{Var}(\beta_p \mid g^X)$ is

$$\mathrm{Var}(\beta_p \mid g^x) = (I + X^T\Sigma^{-1}X)^{-1}.$$

With the posterior distribution in hand, it holds that

$$t(\widehat{x}) = \frac{1}{2}\log(\widehat{x}^T\widehat{x}) - \frac{1}{2}\log(\widehat{x}^T(I + X^T\Sigma^{-1}X)^{-1}\widehat{x}).$$

Besides, the value of data is

$$\mathtt{V}(x, \widehat{x}) = \frac{1}{2}\log(x^Tx) - \frac{1}{2}\log(x^T(I + X^T\Sigma^{-1}X)^{-1}x).$$

Therefore, the extra surplus of reporting $\widehat{x}$ is

$$\mathtt{V}(x, \widehat{x}) - t(\widehat{x}) = \frac{1}{2}\left[\log(x^Tx) - \log(x^T(I + X^T\Sigma^{-1}X)^{-1}x) - \log(\widehat{x}^T\widehat{x}) + \log(\widehat{x}^T(I + X^T\Sigma^{-1}X)^{-1}\widehat{x})\right].$$

In order to satisfy the IC constraint, we need to prove that $t(\widehat{x})$ is a constant for any $\widehat{x}$. However, since the type space is continuous, it holds that $X^T\Sigma^{-1}X$ should be a constant times identity matrix. However, $X$ can be determined by the following optimization problem

$$X = \underset{X}{\operatorname{argmax}} \, \mathbb{E}_x t(x) = \underset{X}{\operatorname{argmax}} \, \mathbb{E}_x \frac{1}{2}\log(x^Tx) - \frac{1}{2}\log(x^T(I + X^T\Sigma^{-1}X)^{-1}x).$$

Therefore, there exists some prior distribution for type $x$, Dirac delta distribution for example, that $X^T\Sigma^{-1}X$ isn't a constant times identity matrix which leads to a contradiction. With the continuity of $\mathtt{V}(\cdot, \cdot)$, there exists a continuous distribution leading to contradiction as well.

Therefore, it means that our assumption that the seller can achieve the first-best revenue is wrong and it ends our proof. $\square$

G.2.2. PROOF OF THEOREM 4.3

For any design matrix $X$, as the proof of Theorem 4.2, the total surplus of the buyer and the seller is no larger than $\mathtt{V}(x, x) = \frac{1}{2}\log(x^Tx) - \frac{1}{2}\log(x^T(I + X^T\Sigma^{-1}X)^{-1}x)$. Therefore, we only need to construct a mechanism satisfying IC and IR constraints that for any type $x$, the seller's revenue is no smaller than $\mathtt{V}(x, x) - \log(\kappa((\sqrt{\Sigma(X)})^{-1}X))$. More generally, we allow the seller to manipulate her data by adding some noise as in Section 4.1 or doing linear combination as in Algorithm 2. However, the data-generating process must be pellucid and known by both the seller and the buyer which is demanded by mechanism design problems. The result can directly lead to Theorem 4.3.

Since $\kappa((\sqrt{\Sigma(X)})^{-1}X)$ is the square root of the condition number or the ratio between the largest and the smallest singular values of $X^T\Sigma(X)^{-1}X$, if $X^T\Sigma(X)^{-1}X$ is not full-rank, Theorem 4.3 is right trivially. So, we only need to consider the situation that $X^T\Sigma(X)^{-1}X$ is full-rank. As $X$ is a $n \times d$ matrix, it means that $n \geq d$ for sure. It means that in order to separate buyers with different personal types, the seller needs at least some amount of data to achieve price discrimination.

For simplicity, we use $X$ to replace $(\sqrt{\Sigma(X)})^{-1}X$. For any dataset, we can transform $(X, Y)$ to $((\sqrt{\Sigma(X)})^{-1}X, (\sqrt{\Sigma(X)})^{-1}Y)$ so that the noise distribution is indeed $\mathcal{N}(0, I)$. It's just for readers' understanding and transforming back leads to the original Theorem 4.3.

The key challenge in our setting is caused by the fact that $x$, the buyer's private type, is continuous. A difficulty caused by continuity is ensuring incentive compatibility. While it is easy to deter buyers from scaling their feature vector when reporting (e.g. rescale all reported features on the seller's end prior to allocation and pricing), stopping buyers from *rotating* the reported feature vector is much harder. As the data design matrix $X$ is not equally informative in all directions in the feature space, by rotating the reported feature vector, the buyer could act as if he did not gain much information from purchasing the data. At the same time, slightly rotating the reported direction could still lead to significant information gain from the buyer due to the small angle between his true private type and his reported type. We visualize the impact of rotating the reported type in Figure 4.

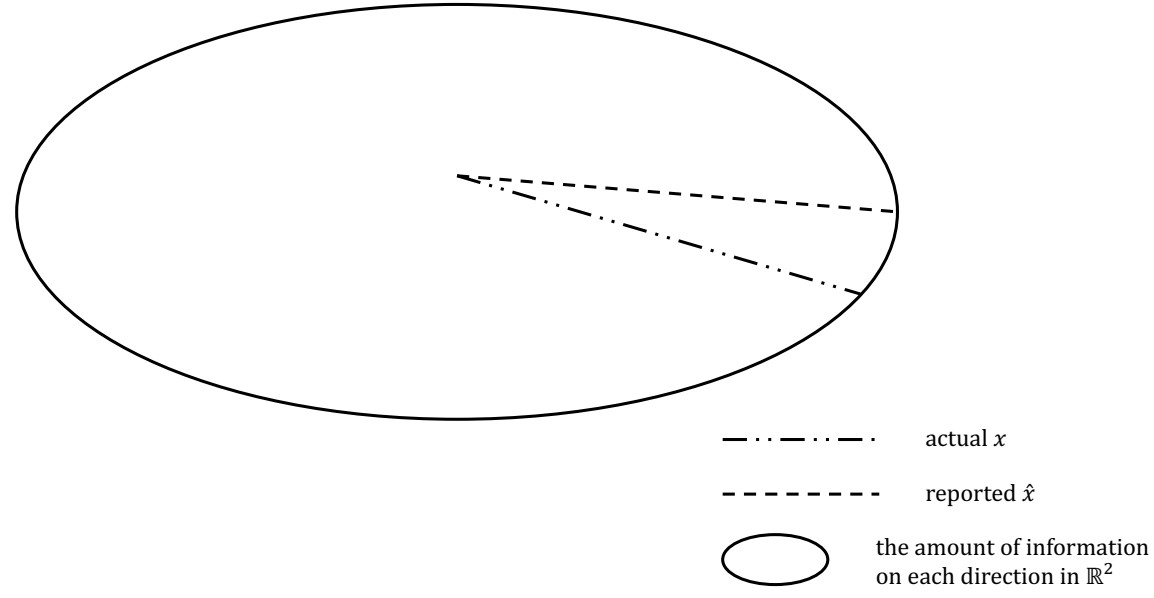

Figure 4. Visualization of the effect of reporting the rotated private type in $\mathbb{R}^2$. The ellipse represents the amount of information that $X$ contains on each direction in $\mathbb{R}^2$. Long-dash-dot-dot line denotes the buyer's private type $x$ and dash line the buyer's report $\widehat{x}$.

Intuitively, the radius of the ellipse in Figure 4 roughly measures the amount of information that $X$ contains on each direction in the parameter space $\mathbb{R}^2$. While the buyer's true type, depicted in long-dash-dot-dot line in Figure 4, benefits the most if he were to report truthfully, by reporting $\widehat{x}$, visualized by the dash line, the buyer could lead the seller to believe that he gains less information from the data purchased, thereby potentially reducing the price the seller charges. Moreover, when the angle between $\widehat{x}$ and $x$ is sufficiently small, the buyer would still gain sufficient information from untruthful reporting, while potentially paying less.

To deter buyers from rotating their report, we propose the SVD mechanism, whose hallmark is performing Singular Value Decomposition (SVD) over the buyer's feature matrix $X$ to obtain

$$X = U \begin{bmatrix} S \\ 0_{(n-d)\times d} \end{bmatrix} V,$$

where $S = \text{diag}\{\lambda_1, \ldots, \lambda_d\}$, $0_{(n-d)\times d} \in \mathbb{R}^{(n-d)\times d}$ is a matrix filled with zeros, and $U \in \mathbb{R}^{n\times n}, V \in \mathbb{R}^{d\times d}$ are unitary matrices. In particular, $\{\lambda_1, .., \lambda_d\}$ are singular values of $X$, we define $X$'s condition number as $\kappa(X) = \frac{\max_i \lambda_i}{\min_i \lambda_i}$. We also

obtain the left-inverse of $X$, $L = U \begin{bmatrix} S^{-1} \\ 0 \end{bmatrix} V$, such that

$$L^T X = V^T \begin{bmatrix} S^{-1} & 0 \end{bmatrix} U^T U \begin{bmatrix} S^{-1} \\ 0 \end{bmatrix} V = I.$$

Geometrically speaking, $U$ and $V$ are rotation matrices that help orthogonalize different components. The entries in the diagonal matrix $S$, $\lambda_i$, are the singular values that dictate the amount of information contained in a specific direction in $\mathbb{R}^d$. Particularly, for the multi-armed bandits setting we deferred to Corollary D.6, $\lambda_i$ equals to $\sqrt{n_i}$ where $n_i$ is the number of observations at the $i$-th component. Intuitively speaking, the larger $\lambda_i$ is, the more information $X$ contains in the particular direction.

Recall that we have the SVD mechanism in Algorithm 2 and let's provide a high-level description of the algorithm. Performing SVD over $X$ recovers the amount of information the seller's data contains on directions in $\mathbb{R}^d$. In particular, these directions are given by the row vectors of $V$, and form an orthonormal basis of the $\mathbb{R}^d$ parameter space. Using the results of SVD, the mechanism constructs projection operator $L$ and normalizes the projection of $\widehat{x}$, $b(\widehat{x})$. The allocation and payment rules are constructed using the projection $b(\widehat{x})$ according to the penultimate line in Algorithm 2.

As the payment rule never charges more than the buyer's information gain whenever he is truthful, the mechanism is individually rational, which we formalize in the following lemma.

**Lemma G.2.** *The mechanism satisfies individual rationality (IR).*

*Proof.* For any truthful buyer, the payment $t(\widehat{x}) = t(x) = \mathtt{V}(x, x)$ is exactly his value of the received data $g[x]$. $\qquad\square$

Proving that the mechanism is incentive-compatible is more involved. Intuitively, projecting the reported type $\widehat{x}$ along $L$ as opposed to $X$ *normalizes* the amount of information gained in each direction. Recall that performing SVD over $X$ yields $U, S^{-1}, V$, from which $L$ is constructed. Previously, we showcased that the diagonal entries in $S$ correspond to the amount of information $X$ has in a particular direction. Conversely, the entries in $S^{-1}$ *normalizes* the amount of information $X$ has along the direction. Projecting $\widehat{x}$ over $L$ then uses the values in $S^{-1}$ to normalize the information the buyer may gain in a particular direction. As opposed to the ellipse we observe in Figure 4, the information $g[\cdot]$ provides for each direction now better resembles a circle, and the buyer is thus discouraged from untruthful reports, as doing so would not alter his payment significantly. We formalize the intuition in the following lemma.

**Lemma G.3.** *The mechanism satisfies incentive compatibility (IC).*

*Proof.* We need to prove that the buyer's utility is maximized when he reports truthfully, namely for all $\widehat{x} \in \mathcal{X}$

$$\mathtt{V}(x, \widehat{x}) - t(\widehat{x}) \leq \mathtt{V}(x, x) - t(x).$$

Recall by construction of $t(\cdot)$ that $\mathtt{V}(x, x) = t(x)$ and $t(\widehat{x}) = \mathtt{V}(\widehat{x}, \widehat{x})$. Thus, we only need to show that $\mathtt{V}(\widehat{x}, x) \leq \mathtt{V}(\widehat{x}, \widehat{x})$. We first note that the payment rule $t(\cdot)$ is scale-invariant. Indeed, for any $\widehat{x} \in \mathcal{X}$ and $c \geq 0$, we have

$$t(c\widehat{x}) = \frac{1}{2} \log(\widehat{x}^T \widehat{x}) + c - \frac{1}{2} \log\left( \widehat{x}^T \left(I^{-1} + X^T b(\widehat{x}) b(\widehat{x})^T X\right)^{-1} \widehat{x} \right) - c = t(\widehat{x}),$$

as we note that $b(\widehat{x})$ is also scale invariant. It is then without loss of generality to assume that $\|x\| = \|\widehat{x}\|$.

Expanding both $\mathtt{V}(\widehat{x}, x)$ and $\mathtt{V}(\widehat{x}, \widehat{x})$, we know that proving IC is equivalent to showing

$$x^T (I + X^T b(\widehat{x}) b(\widehat{x})^T X)^{-1} x \geq \widehat{x}^T (I + X^T b(\widehat{x}) b(\widehat{x})^T X)^{-1} \widehat{x},$$

as we recall the logarithmic function $\log(\cdot)$ is monotonic and note that the matrix $I + X^T b(\widehat{x}) b(\widehat{x})^T X$ is positive definite.

Define the shorthand $f(x, \widehat{x}) = x^T (I + X^T b(\widehat{x}) b(\widehat{x})^T X)^{-1} x$, which is a strongly convex quadratic function. Observe that proving the inequality holds for any pair $x, \widehat{x}$ can be reduced to showing that for any $\widehat{x}$, the induced quadratic function $f(x, \widehat{x})$ is minimized by $\widehat{x}$ over the set $\{x \in \mathbb{R}^d : \|x\| = \|\widehat{x}\|\}$, as we recall it is without loss of generality to focus the set of vectors with the same length as $\widehat{x}$.

By first order condition of strongly convex functions, we need to show that $\frac{\partial f(x,\widehat{x})}{\partial x}\big|_{x=\widehat{x}} = 0$. Since we restrict ourselves to the set $\{x \in \mathbb{R}^d : \|x\| = \|\widehat{x}\|\}$, we only need to prove that $\widehat{x}^T(I + X^T\widehat{x}\widehat{x}^T X)^{-1}y = 0$ for any $y$ such that $\widehat{x}^T y = 0$. By Sherman-Morrison (Lemma G.1), we have

$$(I + X^T b(x)b(x)^T X)^{-1} = I - \frac{X^T b(x)b(x)^T X}{1 + b(x)^T X X^T b(x)}.$$

Clearly $\widehat{x}^T I y = \widehat{x}^T y = 0$. Additionally, we know that

$$\widehat{x}^T X^T b(x)b(x)^T X y = \frac{1}{\|L\widehat{x}\|^2}\widehat{x}X^T L\widehat{x}\widehat{x}^T L^T X y = \frac{1}{\|L\widehat{x}\|^2}\widehat{x}X^T L\widehat{x}\widehat{x}^T y = 0,$$

as $\widehat{x}^T y = 0$ and $L^T X = I$. We then know that

$$\widehat{x} = \operatorname*{argmin}_{x \in \{x \in \mathbb{R}^d : \|x\| = \|\widehat{x}\|\}} f(x, \widehat{x})$$

for any $\widehat{x}$ and consequently, $\mathtt{V}(\widehat{x}, x) \leq \mathtt{V}(\widehat{x}, \widehat{x})$, completing the proof. $\qquad\square$

Finally, we show that the mechanism achieves near-optimal revenue with at most $\log(\kappa(X))$ loss. Note that the maximum revenue is no greater than the social welfare for any IR mechanism. Moreover, selling all data maximizes information gain, and by extension, social welfare. Then, for any private type $x$, a trivial upper bound over the revenue is $\mathtt{V}(g^X; q, x)$, which holds for all possible choices of $g[\cdot]$. Recall that we dub this revenue, $\mathtt{V}(g^X; q, x)$, the first-best revenue, as it is the maximum revenue that any potential non-IC mechanism can extract from a buyer with private type $x$.

We now show that the gap between the revenue achieved by our mechanism and this theoretical upper bound is always no greater than a problem-dependent constant.

**Theorem G.4.** *The SVD mechanism achieves revenue no less than $\mathtt{V}(g^X; q, x) - \log(\kappa(X))$ for any buyer with private feature $x$, where we recall $\kappa(X) = \lambda_{\max}(X)/\lambda_{\min}(X)$ is the ratio between the largest and the smallest singular values of $X$.*

*Proof.* Recall that we use $X$ to replace $(\sqrt{\Sigma(X)})^{-1}X$ for simplicity. By Lemma G.3, our mechanism is truthful, and any buyer with private type $x$ would report truthfully. Recalling the construction of $t(\cdot)$, our goal is to then bound the difference between $\mathtt{V}(x, x)$ and $\mathtt{V}(g^X; q, x)$ which, by definition of $\mathtt{V}(\cdot, \cdot)$, is equivalent to showing

$$\frac{1}{2}\log(x^T(I + X^T b(x)b(x)^T X)^{-1}x) - \frac{1}{2}\log(x^T(I + X^T X)^{-1}x) \leq \log(\kappa(X)). \tag{6}$$

Moreover, examining Inequality (6), we know it is further without loss of generality to assume that $\|x\| = 1$. By construction, $b(x)$ is invariant to the scaling of $x$. The effect of scaling $x$ then cancels out in the left-hand side.

Let us begin with the first term on the left-hand side of Inequality (6). Expanding $(I + X^T b(x)b(x)^T X)^{-1}$ by Sherman-Morrison, we have

$$x^T(I + X^T b(x)b(x)^T X)^{-1}x = 1 - \frac{x^T X^T b(x)b(x)^T X x}{1 + b(x)^T X X^T b(x)}.$$

Focus on the enumerator first. Recalling $b(x) = \frac{Lx}{\|Lx\|}$ where $L$ is the left inverse of $X$, for the enumerator we have

$$x^T X^T b(x)b(x)^T X x = \frac{1}{\|Lx\|^2}x^T X^T Lxx^T L^T X x = \frac{1}{\|Lx\|^2}x^T xx^T x = \frac{1}{\|Lx\|^2}.$$

As for the denominator, again recalling $b(x) = \frac{Lx}{\|Lx\|}$, we have

$$1 + b(x)^T X X^T b(x) = 1 + \frac{1}{\|Lx\|^2}x^T L^T X X^T Lx = \frac{1}{\|Lx\|^2}(\|Lx\|^2 + x^T L^T X X^T Lx)$$

$$= \frac{1}{\|Lx\|^2}x^T(L^T L + I)x.$$

Additionally, note that $L = U \begin{bmatrix} S^{-1} \\ 0 \end{bmatrix} V$ with $U$ and $V$ being orthonormal matrices, we have

$$L^T L = V^T \begin{bmatrix} S^{-1} & 0 \end{bmatrix} U^T U \begin{bmatrix} S^{-1} \\ 0 \end{bmatrix} V = V^T (S^{-1} S^{-1}) V = V^T \text{diag}\left(\frac{1}{\lambda_1^2}, \ldots, \frac{1}{\lambda_d^2}\right) V,$$

as we recall $S = \text{diag}(\lambda_1, \ldots, \lambda_d)$ and is diagonal. We then easily know

$$x^T (L^T L + I) x = x^T V^T \text{diag}\left(\frac{1 + \lambda_1^2}{\lambda_1^2}, \ldots, \frac{1 + \lambda_d^2}{\lambda_d^2}\right) V x$$

and therefore

$$x^T (I + X^T b(x) b(x)^T X)^{-1} x = 1 - \frac{1}{x^T V^T \text{diag}\left(\frac{1+\lambda_1^2}{\lambda_1^2}, \ldots, \frac{1+\lambda_d^2}{\lambda_d^2}\right) V x}. \tag{7}$$

We now turn our attention to $x^T (I + X^T X)^{-1} x$. Applying Woodbury matrix identity (i.e. Lemma G.1), we know $(I + X^T X)^{-1} = I - X^T (I + XX^T)^{-1} X$. Since $X^T (I + XX^T)^{-1} X = V^T \text{diag}\left(\frac{\lambda_1^2}{1+\lambda_1^2}, \ldots, \frac{\lambda_d^2}{1+\lambda_d^2}\right) V$ as we recall the SVD for $X$, we have

$$x^T (I + X^T X)^{-1} x = 1 - x^T V^T \text{diag}\left(\frac{\lambda_1^2}{1 + \lambda_1^2}, \ldots, \frac{\lambda_d^2}{1 + \lambda_d^2}\right) V x. \tag{8}$$

Combining Equation (7) and Equation (8), we have

$$
\begin{aligned}
\frac{x^T (I + X^T b(x) b(x)^T X)^{-1} x}{x^T (I + X^T X)^{-1} x} &= \frac{1}{1 - \sum_{i=1}^d \frac{\lambda_i^2}{1+\lambda_i^2} (Vx)_i^2} \left(1 - \frac{1}{\sum_{i=1}^d \frac{1+\lambda_i^2}{\lambda_i^2} (Vx)_i^2}\right) \\
&= \frac{\sum_{i=1}^d \frac{1}{\lambda_i^2} (Vx)_i^2}{\left(\sum_{i=1}^d \frac{(1+\lambda_i^2)}{\lambda_i^2} (Vx)_i^2\right)\left(\sum_{i=1}^d \frac{1}{1+\lambda_i^2} (Vx)_i^2\right)} \\
&\leq \frac{\sum_{i=1}^d \frac{1}{\lambda_i^2} (Vx)_i^2}{\left(\sum_{i=1}^d \frac{1}{\lambda_i^2} (Vx)_i^2\right)^2} \leq \kappa(X)^2,
\end{aligned}
\tag{9}
$$

where $(Vx)_i$ denotes the $i$-th component of the $d$-dimensional vector $Vx$. Here, the first inequality holds due to the Cauchy-Schwarz inequality while the second inequality holds because the numerator is no larger than $\frac{1}{\min_i \lambda_i^2}$ and the denominator is no less than $\frac{1}{\max_i \lambda_i^2}$. Plugging Equation (9) back into Inequality (6) completes the proof. $\qquad \square$

With the argument that $X$ is full column-rank, it holds that $\kappa(X) = \lambda_{\max}(X)/\lambda_{\min}(X)$ is as same as the square root of the condition number or the ratio between the largest and the smallest singular values of $X^T X$. Therefore, Theorem G.4 leads to the following results that there exists a mechanism that the seller can achieve at least $\mathtt{V}(g^X; q, x) - \log(\kappa(X))$ for any type $x$ which leads to what we need directly. Transforming $X$ back to $\sqrt{\Sigma(X)}^{-1} X$ ends our proof. $\qquad \square$

It means that for any mechanism if the seller's exceptional revenue is less than $\mathbb{E}_x \mathtt{V}(g^X; q, x) - \log(\kappa(\sqrt{\Sigma(X)}^{-1} X))$ or the seller's surplus is larger than $\log(\kappa(\sqrt{\Sigma(X)}^{-1} X))$, the seller will have the motivation to change to the SVD mechanism in order to gain more revenue. Therefore, since the seller is rational, Theorem 4.3 holds.

We conclude Appendix G.2.2 with the following remarks. Firstly, we note that there exists a sharper, yet harder to interpret bound, for Theorem G.4. If we define

$$f(\lambda_1, \ldots, \lambda_d) = \max_y \frac{\sum_{i=1}^d \frac{1}{\lambda_i^2} y_i^2}{\left(\sum_{i=1}^d \frac{1+\lambda_i^2}{\lambda_i^2} y_i^2\right)\left(\sum_{i=1}^d \frac{1}{1+\lambda_i^2} y_i^2\right)},$$

then the SVD mechanism achieves revenue no less than $\mathtt{V}(g^X; q, x) - \frac{1}{2}\log(f(\lambda_1, \ldots, \lambda_d))$ for any buyer with private feature $x$, as Equation (9) is exactly bounded by the function. On the other hand, it means that the buyer can enjoy

only $\frac{1}{2}\log(f(\lambda_1,...,\lambda_d))$ surplus at most. Besides, the condition number of the seller's feature matrix is commonly found in statistical learning literature and is closely related to concentration bounds in the Bayesian linear regression setting (Wasserman, 2004), thus we believe the unfairness between seller and buyers characterized in Theorem 4.3 is easier to interpret.

### G.2.3. PROOF OF COROLLARY 4.4

Now, we are ready to show that the seller can achieve the first-best revenue even if the buyer reports his type later when the data is isotropic.

Since every singular value of $X^T\Sigma(X)^{-1}X$ is the same, it holds that $\kappa(X^T\Sigma(X)^{-1}X) = 1$ with the definition of $\kappa(\cdot)$. Therefore, Theorem 4.3 tells us that the buyer can only enjoy no more than $\log(\kappa(X^T\Sigma(X)^{-1}X))$ surplus which is 0 under this situation. $\square$

This once again demonstrates that the crucial factor is the buyers' willingness to pay, rather than their personal types. Given isotropic data, the seller possesses precise information regarding every buyer's willingness to pay. Accordingly, it is straightforward for her to implement price discrimination, which is manifested by the first-best revenue.

### G.2.4. PROOF OF COROLLARY D.6

To better interpret the proposed SVD mechanism, we show that in the Mult-Armed Bandit (MAB) setting, it reduces to an intuitive mechanism with the first-best revenue. Consider a $d$-armed bandit setting where for any $j \in [d]$, arm $j$'s reward is a random variable independent of other arms' rewards.

The seller's data consists of measurements on each of the $d$-arms and the buyer's private type $x \in \mathbb{R}^d$ is a member of the standard basis of the $d$-dimensional Euclidean space $\mathbb{R}^d$, namely $x \in \mathcal{X} = \{e_1, \ldots, e_d\}$. Similarly, the seller's dataset's features are all members of $d$-dimensional standard basis.

We can thus demonstrate that the SVD mechanism can be simplified to a straightforward and comprehensible mechanism, wherein the seller conveys to the buyer the average reward for the arm in question. It is not arduous to establish that this mechanism attains the first-best revenue, as the average reward constitutes the most accurate estimate for the reward of the buyer's queried arm, given the seller's dataset. We provide a formal articulation of this proposition in the ensuing lemma.

**Lemma G.5.** *In the MAB setting, the SVD mechanism reduces to the following*

$$X[\widehat{x}] = \frac{1}{\sqrt{|\{x_i : x_i = \widehat{x}\}|}} \sum_{d \in \{x_i : x_i = \widehat{x}\}} d, \qquad t(\widehat{x}) = \mathtt{V}(g^{X[\widehat{x}]}; q, \widehat{x}) = \mathtt{V}(\widehat{x}, \widehat{x}), \tag{10}$$

*where we use $X[\cdot]$ to denote the part of the design matrix the seller will keep and the seller forms $g[\cdot]$ correspondingly. Moreover, the mechanism is IC, IR, and achieves the first-best revenue.*

*Proof.* We can manually perform SVD over $X$ to show that in this case, a suitable choice for $V$ is the $d$-dimensional identity matrix $I_d$ and $S = \text{diag}\{\sqrt{|\{x_i : x_i = e_1\}|}, \ldots, \sqrt{|\{x_i : x_i = e_d\}|}\}$, namely, $S$ is the diagonal matrix where each diagonal entry corresponds the number of measurements on that particular arm. Here $U$ is a conforming matrix ensuring that $X = USV$, and the SVD mechanism reduces to Equation (10).

We divide the rest of our proof into three parts, beginning with IR, then IC, and ending with showing that the mechanism achieves the first-best revenue.

**IR** The individual rationality is directly implied by Lemma G.2.

**IC** The incentive compatibility is directly implied by Lemma G.3.

**First-best revenue** We note that

$$\mathtt{V}(g^X; q, x) = h(f_q(x)) - h(f_p(x)|g^X) = h(f_q(x)) - h(f_p(x)|g[x]) = \mathtt{V}(g^{X[x]}; q, x)$$

by chain rule of conditional entropy and the fact that any data point different from $x$ is a measurement of a random variable independent of the buyer's private $y$ which is the label corresponding to $x$, and hence leads to an information gain of zero.

Note that the posterior $f_p(x)$ may come from different datasets and we explicitly give the conditions $X$ and $X[x]$ to make it clarified. From the inequality, we know that for any possible private type $x$, the information gain that the buyer obtains from receiving $g[x]$ equals the information gain that he obtains from observing the entire dataset, $g^X$. Moreover, when the buyer reports his type truthfully, he is charged exactly that amount. Thus, the revenue achieved by the mechanism is the highest among all IR (and potentially non-IC) mechanisms. $\qquad\square$

As Lemma G.5 shows, in the multi-armed bandits (MAB) setting, the SVD mechanism can achieve the first-best revenue which means that the consumer surplus is 0. Therefore, we finish our proof. $\qquad\square$

This highlights a fundamental distinction between the data market and traditional markets for tangible goods. In the data market, the value of a dataset varies significantly among buyers. As in the multi-armed bandit setting, buyers derive no utility from obtaining data about a different arm. Consequently, buyers have no incentive to misreport their valuations, which enables the seller to effortlessly attain the highest possible revenue.

