# OpenReview forum: "An Instrumental Value for Data Production and its Application to Data Pricing"
_ICML.cc/2025/Conference — ICML 2025 poster_

### Official Review · Reviewer_PAUD · 2025-03-13

**Overall Recommendation:** 2

**Summary:**

The paper analyzes how to quantify a datasets' instrumental value by taking into account the prior knowledge/data sources the buyer has. The authors argue that switching from intrinsic value (as in data Shapley) to this instrumental value is useful for avoiding overestimating the data value.

**Claims And Evidence:**

The paper's theory builds on some models that are not very realistic for machine learning applications. But for the studied models, the claims and results seem convincing.

**Essential References Not Discussed:**

N/A

**Experimental Designs Or Analyses:**

There are no experimental results in the paper.

**Methods And Evaluation Criteria:**

- The data generation process described in the paper is a bit too far from what's happening in practice in machine learning. The data sellers typically do not have control over the data they collect, so the perfect customization in the way the authors present seems unrealistic.

- The analysis and its use in estimating data value rely on data generation distribution. However, such a distribution would be very complex and unknown in practice, especially in machine learning.

- The Bayesian regression model oversimplifies the problem without any justification or intuition into why it might be a good model for practical data-utility relation. So, I am not sure how much of the analysis in the paper is actually relevant in practice. Or how straightforward is it to extend to other possible relations?

- The weaknesses above limit the paper's practical relevance to data valuation in machine learning. Perhaps, the authors didn't necessarily think of data used for machine learning. But then, I am not sure if ICML is the right venue for this study.

**Other Comments Or Suggestions:**

-

**Other Strengths And Weaknesses:**

I think it's great to think about the instrumental value of datasets, especially for applications like training foundation models on a mix of many datasets. However, the current analysis makes strong assumptions about the data generation process and how its value would behave, which are not very realistic for machine learning datasets.

**Questions For Authors:**

Do the authors have any ideas on how to extend their results to better suit for machine learning applications?

**Relation To Broader Scientific Literature:**

Due to impractical assumptions and models used in the paper, I don't think the results would be very useful for the machine learning community. They could be more useful in other areas where the assumptions around the data and its value are more realistic.

**Theoretical Claims:**

I checked the theorems and they seem correct to me.

---

> ### Author Rebuttal · Authors · 2025-03-31
>
> We thank the reviewer for the detailed and constructive comments. We will answer your questions according to logical order.
>
>
>
> We respectfully disagree with the reviewer’s comment on relevance to the machine learning community. Below, we provide our detailed explanations:
>
>
>
> 1. In many scenarios, such as experimental design and clinical data, sellers can effectively control the subjects and collect data. Even in the field of deep learning, data annotation companies like Scale AI can provide labeling for data with specific features. These are examples of scenarios where perfect customization can be applied.
>
> 2. Similarly, the Bayesian setting is also widely used in machine learning. For example, please refer to the *Bayesian Clinical Trials* article published in *Nature* (https://www.nature.com/articles/nrd1927). Also, Bayesian updating is also very common in deep learning literature—recent work (https://arxiv.org/pdf/2503.04722) even shows that LLMs exhibit Bayesian behavior.
>
> 3. In Section 2, we consider the general Bayesian setting of valuation, which does not rely on a known data-generating distribution. Then, in Section 3, we examine a special linear case and Gaussian noise. The reasons are as follows: Gaussian noise is the most common type of noise in nature. Regarding linearity, some studies show that data valuation is transferable—the value ranking under linear models often holds under nonlinear ones as well (see right side, lines 253–260). Moreover, when the true model is unknown, linear models are the most robust (lines 65–70). Finally, linearity allows for tractable theoretical analysis, which helps us better understand the underlying economic intuition. Extending our framework and tailoring it to specific scenarios is out of the scope of this paper but a promising direction for future work.
>
> 4. For potential machine learning applications, especially in industries such as clinical healthcare, our instrumental value can effectively evaluate the value of sequential data, whereas the Shapley value tends to overestimate it. Moreover, as shown in Section 2.3, despite certain assumptions possibly being violated in large-scale datasets, instrumental value reduces the computational complexity from exponential to polynomial. Appendix B.3 further demonstrates through numerical experiments that instrumental value exhibits a certain degree of correlation with the widely used Shapley value. Using instrumental value as an empirical approximation for data valuation—and potentially for pricing—in large-scale datasets could be a promising direction for future applications.
>
> 5. ICML features many fundamental machine learning papers, such as those in statistical learning, and is not solely focused on deep learning or empirical work. This is also why the theory track—including areas like game theory—exists. These theoretical contributions provide the foundation and guidance for future large-scale applications. The mutual reinforcement between theory and application is what makes machine learning a broad and inclusive field. Therefore, we believe that the problem we study, categorized in Theory-> Game Theory, is not only of interest to the ML community but also meaningful and impactful.
>
>
>
> We hope our rebuttal has clarified the reviewer’s confusion and respectfully hope that the reviewer would consider re-evaluating the merit of our work accordingly.

---

> > ### Comment · Reviewer_PAUD · 2025-04-04
> >
> > I thank the authors for their response.
> >
> > I see that the authors justify the use of assumptions on data generation distribution and Bayesian regression model, with pointers to a paper on clinical data. Unfortunately, this does not resolve my concerns on practicality for broader machine learning applications. I see that Reviwer TrgW had similar concerns/questions. It is of course fine to focus on a narrow application domain but the paper should be upfront about this limitation. I think the current abstract/introduction is misleading about how generally applicable the results are.
> >
> > The other referenced paper on "LLMs act Bayesian" (https://arxiv.org/pdf/2503.04722) is interesting but I don't see how this is relevant for this paper or the discussion on strong assumptions and Bayesian linear model. The paper is on in-context learning, which, by definition, does not require data beyond the prompts and it's not clear to me how it can justify using Bayesian linear model for LLM data.
> >
> > **About relevance to the ICML:** Thanks for clarifying your perspective on this. The paper seems like a good fit under the Game Theory category. The paper, in its current form, motivates the data valuation problem and the proposed approach for machine learning applications. So, naturally, I expressed my concerns on the practicality of the strong assumptions and the proposed approach (similar to Reviwer TrgW). I am not suggesting including empirical work or focusing on DNNs but the paper should either: (1) make it clear that it targets a narrow application domain (clinical data) and has limitations on how realistic the assumptions are and how practical the methodology is for broader machine learning applications, or (2) explain how the results under these strong assumptions could be extended to other possible data-utility relations.

---

> > > ### Author Response · Authors · 2025-04-06
> > >
> > > Thank you for recognizing that our paper is a good fit for the Game Theory track.
> > >
> > > In Section 2, we consider **the most general form of Bayesian regression (without the linearity assumption)** and introduce the concepts of **valid value functions** and cost of uncertainty, along with their characterization (Proposition 2.6). In Section 3, we then focus on linear Bayesian regression, where we derive closed-form solutions and explore a mechanism design problem in this setting. We acknowledge that the linearity assumption imposes certain limitations on the applicability of our results. However, we would like to emphasize that in specific scenarios—such as clinical data, which represents a large and important market—this assumption is realistic and not restrictive in practice. **Moreover, for broader machine learning contexts, Section 2 rigorously defines what constitutes a valid valuation, which is independent of the linearity assumption**. While we recognize the technical challenges posed by model complexity, our goal with Section 2 is to provide economic insights and to use our foundational analysis to spark further interest in the emerging field of data pricing—particularly relevant in the current AI era. For example, companies like Scale AI have shown significant interest in data pricing.
> > >
> > > Specifically, we referred to https://arxiv.org/pdf/2503.04722 to emphasize that the Bayesian approach is a common modeling framework in machine learning, just like the frequentist approach. **As for the linear assumption, it simplifies our model theoretically, making analysis tractable**. Moreover, linear models are known to be robust [1] and transferable[2,3], even in the presence of model misspecification.
> > >
> > > [1] Besbes, O. and Zeevi, A. On the (surprising) sufficiency of linear models for dynamic pricing with demand learning. Management Science, 61(4):723–739, 2015.
> > >
> > > [2] Jia, R., Wu, F., Sun, X., Xu, J., Dao, D., Kailkhura, B., Zhang, C., Li, B., and Song, D. Scalability vs. utility: Do we have to sacrifice one for the other in data importance quantification? In Proceedings of the IEEE/CVF Conference on Computer Vision and Pattern Recognition, pp. 8239–8247, 2021.
> > >
> > > [3] Schoch, S., Xu, H., and Ji, Y. Cs-shapley: Class-wise shapley values for data valuation in classification. arXiv preprint arXiv:2211.06800, 2022.
> > >
> > > Thank you for your suggestion on clearly stating the scope of our paper. **In the camera-ready version, we will make it explicit that Section 2 provides a pricing guideline for a general class of models. The subsequent sections introduce stronger assumptions, and we will clarify in which domains these assumptions are natural, as well as the challenges and potential of removing them**. This will extend the discussion currently found on the right side of lines 258–262 and lines 425-431.

---

### Official Review · Reviewer_84M6 · 2025-03-15

**Overall Recommendation:** 3

**Summary:**

The paper studies the mechanism design problem in pricing and designing data generating processes in the context of Bayesian regression. Concretely, the buyer first has a prior $q$ of the regression parameter, reports his feature $x$ that he wants prediction on, and then upon obtaining data (including the data points $D$ and knowledge of the data generating process $g$) at some price $t$, forms a posterior $q$. Buyer's utility in the setup is the Bregman divergence between p and q corresponding to the entropy cost. The paper consider the mechanism design problem of (D,g,t), such that certain nice mechanism design properties are achieved, such as IC, IR, and regret. The paper separately considers perfect customization and limited customization settings, the second one being the harder setting to analyze.

**Claims And Evidence:**

All claims are supported by proofs.

**Essential References Not Discussed:**

The paper cited sufficient related works.

**Experimental Designs Or Analyses:**

The main text only contains theories.

**Methods And Evaluation Criteria:**

The paper discusses classical mechanism properties such as IC, IR and regret.

**Other Comments Or Suggestions:**

Some comments on writing.
1. In terms of writing, the main paper only really considers the entropy setting, while the section 2 discusses settings of a general cost functions. The Characterization of Valid Valuation Functions is interesting, but for streamlining the paper I feel the main mechanism design problem should maybe be described first in a self-contained way.

2. In example 2 the derivation of the variance is missing. I assume the computation of posterior is omitted?



Line 275 notably.

Nortation $g^n$ in Theorem 4.1 needs to be explained near the theorem.

**Other Strengths And Weaknesses:**

Strengths

1. The problem of pricing additional data nowadays is crucial given that large language models require new data to perform alignment/ fine-tuning/RAG etc. The paper provides a theoretical discussion of pricing and mechanism design of incremental value of data in the Bayesian regression setting and devises several mechanisms.

2. Overall the paper is well written and provides sufficient explanation of concepts. I also like Example 1 and 2 to explain the setup of selling data generating processes and changes in variances.

3. The paper contains a few interesting results for pricing data, such as the equivalence between valid valuation functions and Bregman divergence, and also the results that in the limited customization setting, there is not 0-regret generally but if the design matrix is isotropic then there is.

Weakness

To me there are a few clarifying problems to be addressed to make the problem setup clearer, eg concretely what are exactly known to buyer when the mechanism is announced (see below). Based on the above I recommend weak accept, and I'm happy to adjust the score if my concerns are addressed.

**Questions For Authors:**

A few clarifying questions.

1. What is the corresponding utility function for the valuation function in Theorem 3.1?
2. In the limited customization setting, are the design points also announced as to the buyer?
3. In section 2.1 it says "This will be useful for selling DPPs since it means the value of a DPP can be quantified without seeing the realized data". But in section 4 the seller has a variance function $\sigma$. So is $\sigma$ known to buyer?

**Relation To Broader Scientific Literature:**

The paper cited sufficient related works.

**Theoretical Claims:**

I checked the proof of Theorem 4.1 which relies on a close-form expression of the valuation function and a generalization of results from Myerson saying that the pricing rule takes the form of derivative of the valuation.

---

> ### Author Rebuttal · Authors · 2025-03-31
>
> Thank you for your kind remarks and questions, and we would now like to answer your questions in logic order. Below, we first address the reviewer's major questions, and then clarify a few minor comments. We are happy to engage with any further questions.
>
>
>
> **Re Corresponding Utility Function in Thm 3.1**: In Theorem 3.1, the corresponding utility function reflects the confidence in the estimation, which is captured by the entropy. As our confidence in the estimation increases, the corresponding entropy decreases.
>
>
>
> **Re Design Matrix in the Limited Customization**: In the limited customization setting, we assume that the design matrix $X$ is public knowledge, but the buyer does not observe the corresponding responses $Y$. In practice, for example, in clinical trials, information about the subject group is usually known, but the evaluation of the drug’s effectiveness needs to be purchased.
>
>
>
> **Re Realized Data and Noise**: By "realized data," we refer to the realized response $Y$, while the variance function $\sigma$ is assumed to be known. In real-world scenarios, sellers typically disclose the measurement error or the specifications of the experimental instruments, such as their precision. When the buyer has no information about data quality, pricing in our setting becomes infeasible.
>
>
>
>  **Re Other Comments Or Suggestions:**
>
> 1. Thanks for your suggestion. Our idea is to first introduce the characteristics of valid valuations (Sections 2 and 3), and then present a concrete application from game theory, say, mechanism design (Section 4). We'll include a roadmap in the final version to outline the content of each section.
> 2. Due to the page limit, we have deferred the calculation of the posterior distribution and variance to the appendix. Please refer to Lines 667–677.
> 3. Thank you for pointing out the typo. We will correct it in the camera-ready version.
> 4. We introduced the definition of $g^n[x]$ in Lines 334–343, which produces $n$ responses for the buyer’s type $x$. Thank you for your suggestion — we will highlight its definition again around Theorem 4.1.
>
>
>
> We hope our rebuttal has clarified your concerns and would greatly appreciate your re-evaluation of our work’s merits if these clarifications address your doubts.

---

### Official Review · Reviewer_TrgW · 2025-03-16

**Overall Recommendation:** 2

**Summary:**

This paper introduces a framework for quantifying the instrumental value of data production processes (DPPs) under the bayesian linear model. The authors focus on how much additional benefit (or marginal contribution) new data brings to a decision‐maker’s task. The proposed data value is mathematically equivalent to information gain. They then leverage this valuation to study optimal pricing in data markets. Two key selling scenarios are explored: (a) perfect customization—where the seller can generate data tailored exactly to the buyer’s needs, allowing for full (first-best) surplus extraction; and (b) limited customization—where the seller can only curate data from an existing pool, yielding a mechanism that, while not fully optimal, achieves revenue within a bounded regret.

**Claims And Evidence:**

**Claims:**

Instrumental Value Framework: It argues that the value of data should be measured in terms of its marginal improvement on a decision-maker’s utility rather than by an average contribution (as in Data Shapley).

Microfoundations via Bayesian Decision-Making: The authors claim that by grounding the valuation in a contextual Bayesian decision-making problem, one can rigorously justify the use of Bregman divergence to capture data’s value.

Optimal Pricing Mechanisms: For perfect customization, they claim that there exists an incentive-compatible and individually rational mechanism that extracts full surplus (zero-regret). Under limited customization, they provide a mechanism based on singular value decomposition (SVD) that achieves revenue nearly as high as the first-best benchmark, with regret bounded by a term related to the condition number of the data matrix.

**Evidence:**

The evidence provided is largely theoretical under strong assumptions such as Bayesian Linear Model and . The authors back their claims through a series of definitions, propositions, and theorems (e.g., Theorem 3.1, Theorem 4.1, and Theorem 4.3), complete with mathematical derivations. Although the paper includes references to numerical experiments in an appendix B.3.1, the investigation is mainly showing it is faster and close to DataShapley, while there are existing works to speed up shapley value calculation (e.g., https://arxiv.org/abs/2107.07436) which the authors did not consider.

I find it weird that the only numerical evaluation is to demonstrate the instrumental value is close to data shapley and there is no numerical evidence that the instrumental value is a better metric.

**Essential References Not Discussed:**

The authors did not cite most data attribution methods after 2022, which seems neglected many related works. Also, I think most these works study values similar to the ``instrumental value'' that considers additional knowledge gains (https://arxiv.org/abs/2405.13954).

Active learning is a widely-studied field that measures the value of additional data. It is very established that information gain can be used to quantify the additional ``instrumental value'' of new data. The paper did not cite active learning papers.

**Experimental Designs Or Analyses:**

See my comments on numerical analysis above.

**Methods And Evaluation Criteria:**

The methodology is primarily theoretical:

* Framework Development: The authors define a data production process (DPP) and introduce a valuation function that depends on the buyer’s prior beliefs and decision context.

* Microfoundations: By invoking a contextual Bayesian decision-making framework, the paper ties data valuation to improvements in expected utility. I have concerns on the strong assumption of the bayesian linear model and the DPP assumption. I find the data shapley's framework that replies on the realized data much easier to understand. In practice, it is quite difficult for the data buyer to formulate their prior belief and how their belief will be updated after acquiring the data. If the practical scenario is more complex than a simple bayesian linear regression update, how should I apply this paper's method and why it is better than data shapley? I think the authors should include more realistic examples to strengthen the paper.

* the numerical experiments in B.3.1 only shows the instrumental value is comparable to data shapley, which I don't think bring much value to the paper. Ideally I expect to see how in practice the proposed method should be used and why it is better than the data shapley.

**Other Comments Or Suggestions:**

NA.

**Other Strengths And Weaknesses:**

NA.

**Questions For Authors:**

NA.

**Relation To Broader Scientific Literature:**

The paper is closely related to data shapley and data attribution methods.

I think most methods in the literature focus on the instrumental value, so the claims that it is novel is weak. The claim that the data buyer can update prior knowledge part I feel is widely studied in theoretical work such as information sharing between retailers with the similar theoretical framework that authors did not discuss (e.g., https://pubsonline.informs.org/doi/10.1287/msom.2020.0915).

**Theoretical Claims:**

Overall the proofs seem correct. I have some issues:

*  it seems the authors are considering the data utility and the customer valuation as the same thing, but I do not think they are equivalent. The valuation describes how much the data can improve the posterior update, but valuation describes how much the customer is willing to pay in the monetary value. How can the authors find the mapping in practice? The paper should also be significantly changed in terms of this.

* I think the whole theoretical proof until theorem 3.1 is a re-invention of bayesian active learning in the name of ``instrumental value''. The value of data is commonly defined as information gain in the active learning literatures. There is nothing new here. The paper also did not cite any active learning papers.

---

> ### Author Rebuttal · Authors · 2025-03-31
>
> We thank the reviewer for the detailed and constructive comments. We will answer your questions according to logical order.
>
>
>
> **Comparison with Other Data Valuation Methods**: Compared to the fairness-based Shapley value, our instrumental value accounts for sequential data settings where Shapley can underestimate value (cf. Example 4). Theoretically, instrumental value avoids the exponential complexity of Shapley, making it more scalable for large ML datasets. The acceleration methods you mentioned just approximate the Shapley value, and similar ideas may apply to instrumental value. Our experiments show the conceptually significant advantages of instrumental value in terms of computation and sequential settings. Defining a universally better metric is unrealistic and not our intention. Meanwhile, Shapley value in fact also requires a suitable choice of valuation function, while we, under the Bayesian framework, rigorously characterize what valuations are valid for decision making (Sec 2).
>
>
>
> As for https://arxiv.org/abs/2405.13954, it also stems from Shapley's inefficiency but takes a different angle—focusing on gradient-based LLMs. In contrast, we take an economics perspective (hence the game theory track), aiming for decision-oriented valuation (Sec 2.2) with theoretical guarantees, which they do not pursue. Thanks for your suggestions—we’ll discuss these works in the final version.
>
>
>
> **Discussing Our Assumptions**: Regarding the Bayesian assumption, we can use either strong or uninformative beliefs (see right, lines 120–130), just to cover the general case. Bayesian updating is also very common—recent work (https://arxiv.org/pdf/2503.04722) even shows that LLMs exhibit Bayesian behavior. The paper you mentioned (https://pubsonline.informs.org/doi/10.1287/msom.2020.0915) also adopts a standard Bayesian setting, but we focus on what constitutes a valid valuation under this framework (Sec 2), while it assumes a specific utility function. We then explore the mechanism design problem under valid valuations, which is also entirely new.
>
>
>
> Regarding linearity, some studies show that data valuation is transferable—value order under linear models often holds under non-linear ones as well (see right side, lines 253–260). Moreover, when the true model is unknown, linear models are the most robust (lines 65–70). Finally, linearity enables tractable theoretical results, which help us analyze the underlying economic intuition.
>
>
>
> As for the DPP assumption, one example is mechanism design in clinical trials, where experimenters actively select samples and often update beliefs in a Bayesian manner—see *Bayesian Clinical Trials* published in *Nature* (https://www.nature.com/articles/nrd1927). Building an experimental platform based on instrumental value is a promising direction with high scientific and economic potential.
>
>
>
> **Utility vs Valuation**: Our data valuation measures how much utility the posterior distribution (after observing data) brings over the prior—for example, more precise estimates (with lower variance or entropy) yield higher monetary revenue. This mapping is common in practice; in clinical trials, for instance, pharmaceutical companies often have clear estimates of the value of improved precision.
>
>
>
> **Difference from Active Learning**: While data sequence has effects on both our instrumental data value and active learning, however these two areas are completely different. Active learning studies how to select sequential data and train models to maximize **accuracy**, whereas our paper studies how to define a valid data valuation function that captures downstream users’ utilities. We approach this study from a utilitarian perspective, and our value corresponds to information gain only at a very canonic special case. We will clarify this point in the camera-ready version.
>
>
>
> We hope our rebuttal has clarified the reviewer’s confusion and respectfully hope that the reviewer would consider re-evaluating the merit of our work accordingly.

---

### Decision · Program_Chairs · 2025-05-01

**Decision:**

Accept (poster)

**Comment:**

This paper introduces a framework to quantify the instrumental value of data within a Bayesian linear model. The main idea is that the value of data should be measured in terms of its sequentially updated utility instead of using a static notion such as Shapley value. Based on the instrumental value, the authors propose mechanisms for data pricing in two scenarios: Perfect customization and Limited customization. The main results are: (1) The instrumental value naturally captures the information gain in a Bayesian setting, and (2) When designing a data-pricing mechanism under these assumptions, one can achieve high revenues with appropriate incentive-compatible rules.

This paper is clearly motivated and delivered, with theoretical claims rigorously proved in the appendix. The concepts of mechanism design integration such as IC and IR are crucial, and is also novel in the ML community. Since there contains the category of “game theory”, I think ICML should be a suitable value for this paper as they are solving ML problems (i.e. data evaluation from the perspective of sequential information gain).

However, there are still a few concerns among the reviewing boards. Firstly, whether the instrumental value is significantly different from Shapley value is unclear and lack of evidence. As pointed out by reviewers, the authors should provide more empirical results on practically how we should choose which metric to use. Secondly, the technical contributions main originate from Bayesian active learning, which has been well established. Although the authors claim that the goal of active learning differs a lot with data evaluation, the concepts and methodologies are similar in some sense.

We kindly suggest the authors expand numerical demonstrations on the necessity of using instrumental value instead of Shapley and/or other notions. Besides, the difference between Bayesian active learning (and also the necessity of using Bayesian approach) should be stated more clearly in the next version.